# LEAVE NO OBSERVATION BEHIND: REAL-TIME CORRECTION FOR VLA ACTION CHUNKS

## ABSTRACT

To improve efficiency and temporal coherence, Vision-Language-Action (VLA) models often predict action chunks; however, this action chunking harms reactivity under inference delay and long horizons. We introduce *Asynchronous Action Chunk Correction (A2C2)*, which is a lightweight real-time chunk correction head that runs every control step and adds a time-aware correction to any off-the-shelf VLA's action chunk. The module combines the latest observation, the predicted action from VLA (base action), a positional feature that encodes the index of the base action within the chunk, and some features from the base policy, then outputs a per-step correction. This preserves the base model's competence while restoring closed-loop responsiveness. The approach requires no retraining of the base policy and is orthogonal to asynchronous execution schemes such as Real Time Chunking (RTC). On the dynamic KINETIX task suite (12 tasks) and LIBERO SPATIAL, our method yields consistent success rate improvements across increasing delays and execution horizons ($+23\%$ point and $+7\%$ point respectively, compared to RTC), and also improves robustness for long horizons even with zero injected delay. Since the correction head is small and fast, there is minimal overhead compared to the inference of large VLA models. These results indicate that A2C2 is an effective, plug-in mechanism for deploying high-capacity chunking policies in real-time control.

## 1 INTRODUCTION

Recent advances in large vision–language–action (VLA) models have significantly expanded the capability of robots to generalize across tasks and environments (Black et al., 2024; Gemini Robotics Team et al., 2025; NVIDIA et al., 2025; TRI LBM Team et al., 2025). However, large model size requires high computational cost to output the actions for each step, which leads to high inference latency (Kawaharazuka et al., 2025; Black et al., 2025). Especially in dynamic control, such delays become critical. A robot relying on long action sequences predicted from outdated observations can drift, overlook cues, or fail in tasks demanding rapid reactions, such as catching moving objects or stabilizing unstable systems.

The trend of scaling up neural network policies using foundation models brings representational benefits (Sartor & Thompson, 2025), but also incurs a latency problem. For instance, large VLA models such as $\pi_0$ (Black et al., 2024) or OpenVLA (Kim et al., 2024) have billions of parameters and often require hundreds of milliseconds to generate a single action chunk. These action chunks are predicted solely from the previous observation and then executed in an open-loop manner, without incorporating new sensory input during their execution. In addition, latency not only delays execution but also prevents the policy from incorporating the latest observations, thereby weakening its ability to produce reactive behaviors. This is particularly problematic in tasks where the environment changes rapidly during inference. For instance, following a moving object on a cluttered table or grasping a utensil while other objects are being placed, the robot should adjust its action sequence to new sensory inputs. In these scenarios, actions computed from outdated observations accumulate errors over time, which lowers success rates and, in some cases, leads to task failure. This is the central challenge we address in this work.

Conventional approaches attempt to mitigate the latency of large models through action chunking (Zhao et al., 2023; Black et al., 2024). By predicting long sequences of actions at once, these methods reduce the frequency of expensive inference calls. However, the chunking strategies can impact performance; robots may experience waiting time during inference, and inconsistencies can arise

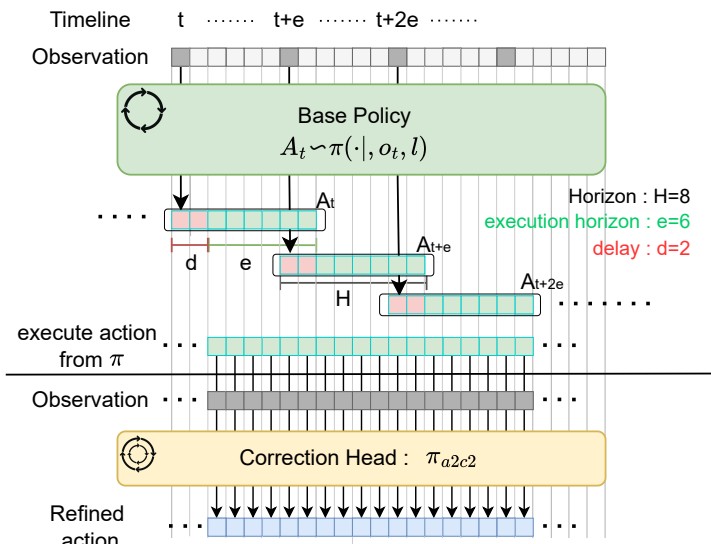

Figure 1: Overview of the proposed A2C2 method. The Base Policy predicts an action chunk with a horizon length of $H$. Due to an inference latency of $d$ steps, the initial $d$ steps of the chunk are discarded. The subsequent $e$ steps are used for the execution horizon. To ensure precision, the Correction Head refines these actions by conditioning on the latest observations.

between successive chunks (Liu et al., 2025). To address this, SmolVLA (Shukor et al., 2025) introduces synchronous execution of the policy, and Real Time Chunking (RTC) (Black et al., 2025) ensures smoother continuity between chunks under asynchrony for diffusion-based action generation. However, these methods still assume that the model predicts fixed-length horizons, which means reactivity to new sensory input remains limited.

Another line of work adopts hierarchical architectures inspired by dual-system reasoning (Kahneman, 2011). Large models serve as a high-level planner (System 2), while smaller policies act as fast executors (System 1). Examples include Hi Robot (Shi et al., 2025), which combines a VLM at the high level with a VLA at the low level, and GR00T-N1 (NVIDIA et al., 2025), which uses a compact policy to refine continuous action chunks. However, since the low-level executor has to wait for predictions from the high-level model, the latency still persists. Consequently, while chunking and hierarchical approaches alleviate some issues, they do not fundamentally solve the challenge of maintaining responsiveness to new observations under the inference delays inherent to VLAs with a large number of parameters.

To mitigate this problem, in this paper, we propose *Asynchronous Action Chunk Correction (A2C2)*, which is a lightweight correction head that can be executed at every timestep to complement the outputs of large VLA models. Unlike conventional approaches such as action chunking and asynchronous inference, our method introduces a lower-level correction layer that directly integrates the most recent observation referring to the action chunks that high-level model outputs. This correction head does not compete with base (high-level) policies like diffusion- or VLA-based chunk generators; instead, it enhances them by injecting real-time feedback to maintain responsiveness under inference delays and long horizons. Through this design, the proposed framework achieves robustness against dynamic environmental changes and external disturbances, thereby mitigating the critical latency bottleneck in deploying large-scale VLA models for real-time robotic control.

In our experiments on the Kinetix tasks, we measure a $35\%$ point increase in success rate over naive execution and $23\%$ point increase over RTC in the presence of delay. For long execution horizons, we measure a $12\%$ point success rate increase over naive execution and $7\%$ point increase over RTC.

In summary, the contributions of this work are as follows:

- We first formulated delays in policy inference with VLAs that generate action chunks.

- A lightweight add-on action correction policy (*A2C2*) is introduced to improve reactivity, which can be applied to any VLA model independent of the underlying architecture.

- The method showed substantial improvements in success rates on dynamic tasks and robot manipulation benchmarks with varied inference delays.

## 2 PROBLEM FORMULATION

We consider an action chunk execution with an imitation learning (IL) policy. As illustrated in Figure 1, an action chunk $A_t = \{a_t, \ldots, a_{t+H-1}\}$ is from IL policy $\pi$ based on the observation $o_t$ and a language instruction $l$. $H$ is the horizon length, the training sequence length of the IL model $\pi$. We assume it uses $e$ steps of the action chunk, and define it as the execution horizon. Policy predicts the action chunk every $e$ steps as follows:

$$A_t = \{a_t, \ldots, a_{t+H-1}\} = \pi(o_t, l). \tag{1}$$

Also, there is an inference latency. We define the *delay* $d$ as the number of control steps between receiving an observation $o_t$ and obtaining the corresponding action chunk $A_t$. Formally, it is computed as

$$d = \left\lfloor \frac{\delta}{\Delta t} \right\rfloor, \tag{2}$$

where $\delta$ represents the combined inference and communication time, and $\Delta t$ denotes the duration of a single control step.

To control delayed, chunked action execution, the agent executes one action per step till a new chunk arrives asynchronously. Additionally, we assume that the policy server can handle only one inference at a time. If the execution horizon $e$ is shorter than the delay $d$, there will be no action during the model inference, which leads to waiting time. On the other hand, if the execution horizon $e$ is longer than $H - d$, there is no action remaining during the inference time. Therefore, the execution horizon $e$ needs to be longer than the delay $d$, and $e$ must be shorter than $H - d$ ($d \leq e \leq H - d$).

In this setting, the agent needs to use the actions that are always based on past observations. Each executed action corresponds to an observation at least $d$ steps old. And in the worst case, the agent may need to execute an action that is generated from the $d + e$ steps past observations.

## 3 METHOD

### 3.1 OVERVIEW

We extend the action chunk–based policy $\pi$ by *Asynchronous Action Chunking Correction (A2C2)*, introducing a lightweight **correction head** $\pi_{a2c2}$ that refines each action within a predicted chunk using the most recent observation, features of the base policy, and a temporal position feature. This framework enables step-wise online correction without retraining the base policy and is complementary to methods such as RTC (Black et al., 2025).

At time t, Observation $o_t$ is sent to the policy server. Then, the base policy $\pi$ generates the action chunk $A_t = \{a_t^{\text{base}}, \ldots, a_{t+H-1}^{\text{base}}\}$ within inference delay $d$ as

$$A_t = \{a_t^{\text{base}}, \ldots, a_{t+H-1}^{\text{base}}\} = \pi(o_t, l). \tag{3}$$

Subsequently, at time $t+k$ ($d \leq k \leq d+e$), time feature $\tau_k$, and base action $a_{t+k}$, latest observation $o_{t+k}$, base policy latent representation $z_t$ and language instruction $l$ are added to the correction head $\pi_{a2c2}$. Since the base policy is inferred only at the beginning of the chunk (time $t$), the latent representation $z_t$ remains constant for all correction steps $k$ within the chunk. The positional feature $\tau_k$ is represented by a sinusoidal embedding that provides periodic structure over the chunk length $\left(\sin(2\pi \frac{k}{H}), \cos(2\pi \frac{k}{H})\right)$. The correction head integrates this information and predicts the residual action $\Delta a_{t+k}$ as

$$\Delta a_{t+k} = \pi_{a2c2}(o_{t+k}, a_{t+k}^{\text{base}}, \tau_k, z_t, l). \tag{4}$$

The residual action $\Delta a_{t+k}$ is added to the base action $a_{t+k}$ and output the execution action $a_t^{\text{exec}}$ as

$$a_{t+k}^{\text{exec}} = a_{t+k}^{\text{base}} + \Delta a_{t+k}. \tag{5}$$

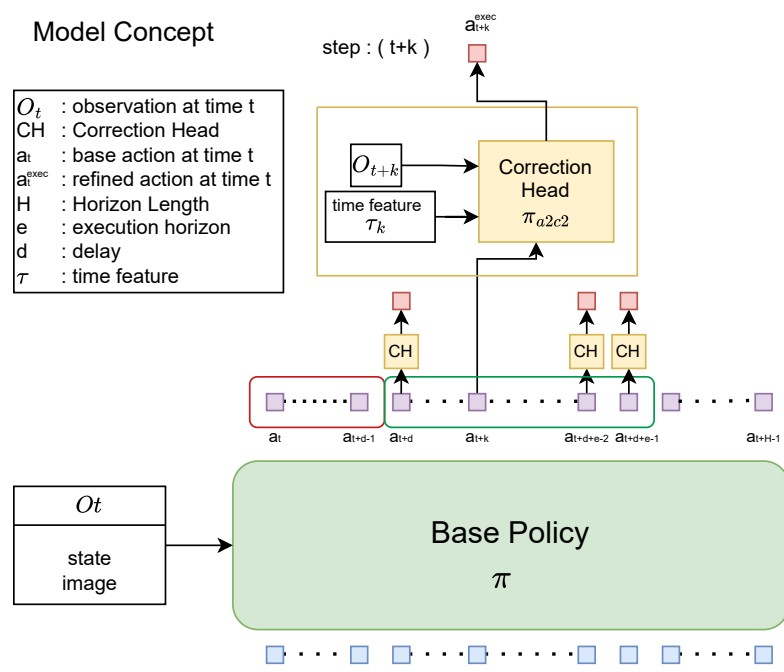

Figure 2: The Base policy $\pi$ outputs an action chunk $A_t = \{a_t, \ldots, a_{t+H-1}\}$ from the current observation $o_t$. For each step within the chunk, a lightweight correction head refines the corresponding base action $a_{t+k}$ using the latest observation $O_{t+k}$ and a time feature $\tau_k$ indicating the relative position within the chunk. The refined actions $a_{t+k}^{exec}$ mitigate performance degradation under inference delays $d$ and long horizons $H$.

Base policy $\pi$ infers an action chunk every $e$ steps with $d$ delay. On the other hand, we assume that the model size of the correction head $\pi_{a2c2}$ is small enough to run every step, which means the inference time of the head is smaller than the duration of a single control step $\Delta t$. Refer to Figure 2 for the overview.

Our method differs from existing approaches for asynchronous inference in the following aspects:

- **Time-aware correction:** The correction head explicitly conditions on the position within the action chunking VLA using a temporal feature.

- **Chunk-level smoothness:** By specifying which element of the chunk is being corrected, the method produces smoother corrections across horizons.

- **Data compatibility:** Training uses the same demonstration datasets as the base VLA policy, which does not require reinforcement learning fine-tuning.

- **Real-time feedback:** New observations are always incorporated, improving robustness under inference delay in dynamic tasks.

## 3.2 MODEL TRAINING PROCEDURE

First, we train the base policy $\pi$ with the dataset

$$D_{base} = \{\{\{o_t, a_t\}_{t=0\ldots T_n}^n, l^n\}_{n=1\ldots N}\},\tag{6}$$

where $N$ denotes the number of episodes in the dataset. Afterward, we add the output action chunk $\hat{A}_t$ of the inference from base policy $\pi$ for each step in the dataset $D_{base}$ as

$$\hat{A}_t = \{\hat{a}_t, \ldots, \hat{a}_{t+H-1}\} = \pi(o_t, l).\tag{7}$$

With these inference results, we created a new dataset for correction head training $D_{cor}$ as

$$D_{cor} = \{\{\{o_t, a_t, \hat{a}_{t-k}^k, \tau_k\}_{t=0\ldots T_0, k=0\le k\le min(t,H-1)}^n, l^n\}_{n=1\ldots N}\}.\tag{8}$$

$\hat{a}_{t-k}^k$ is the k-th action in the action chunk inferred by the base policy from the observation at time $t - k$. Then, the Correction head $\pi_{a2c2}$ is trained to predict the residual action, i.e., the difference between the target action and the base policy output. The target action is the action in the dataset that was originally collected from expert demonstrations. Formally, given the target action $a_{\text{target}}$ and the base policy output $a_{\text{base}}$, the residual target is defined as

$$\Delta a_{\text{residual}} = a - \hat{a}.$$

$\hat{a}$ is a base action inferred by the base policy. There are some possible combinations of the base action with different time features $\tau$. The predicted residual action is denoted by $\Delta a_{\text{residual}}$. The loss function is the mean squared error (MSE):

$$\mathcal{L}_{\text{MSE}} = \frac{1}{N} \sum_{i=1}^{N} \left\| \Delta a_{\text{residual}}^{(i)} - \left( a^{(i)} - \hat{a}^{(i)} \right) \right\|_2^2.$$

Where $N$ denotes the batch size, i.e., the number of training samples in a mini-batch.

## 4 EXPERIMENTAL SETUP

### 4.1 BENCHMARK AND DATASETS

We use the two simulation environments, Kinetix and LIBERO Spatial, for the experiments. Kinetix is first used for evaluating the performance under highly dynamic manipulation and locomotion tasks. Secondly, we used the LIBERO Spatial benchmark to evaluate the performance as a standard benchmark of robot manipulation. Especially, because Shukor et al. (2025) reports that long-horizon significantly degrades performance in LIBERO Spatial, making the task a natural choice for evaluating robustness under long horizons.

Also, for our real world evaluation, we utilize the SO-101 robotic arm. We select a simple pick and place task for the experiments, which is similar to the LIBERO Spatial Benchmark.

### 4.1.1 KINETIX

We used the **Kinetix**, which provides demonstrations across 12 highly dynamic (see Appendix A.1 ). It includes environments ranging from locomotion and grasping to game-like settings. Importantly for our setting, Kinetix contains highly dynamic environments where delayed or inconsistent action generation quickly leads to failure. This makes it a natural testbed for studying the limitations of action chunking and for benchmarking inference-time algorithms such as RTC, which aim to preserve responsiveness and continuity under latency.

Unlike quasi-static benchmarks, Kinetix environments employ torque- and force-based actuation, making asynchronous inference crucial. Kinetix consists of 12 tasks without language input. 1 million steps data was collected by using expert model. Following RTC experiments, we first train expert policies using RPO (Rahman & Xue, 2022) and a binary success reward. For each environment, 1-million transition dataset is generated with the expert policy.

### 4.1.2 LIBERO

LIBERO is a benchmark suite designed to study lifelong robot learning with a focus on knowledge transfer across tasks (Liu et al., 2023). They offer several task suites and datasets. In this work, we specifically use the LIBERO Spatial dataset, which emphasizes spatial reasoning in manipulation tasks as a widely used benchmark for robot manipulation.

For benchmarking 3D robot manipulation, we used **LIBERO spatial** benchmark, which provides 432 episodes and 52,970 frames across 10 tasks. The dataset consists of multimodal input, including top and wrist RGB images ($256 \times 256$), an 8-dimensional state, and language instructions.

### 4.1.3 REAL WORLD TASK

We utilize the SO-101 robotic arm as the experimental platform for conducting our real-world evaluations. For benchmarking the proposed methodology, we selected a simple pick-and-place task. Specifically, the agent is required to grasp a designated object (e.g., a yellow cube) and accurately place it into a specified container (e.g., a bowl), a setup which is designed to be analogous to tasks

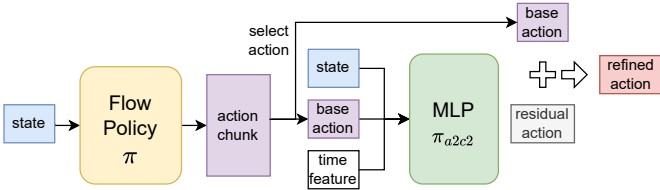

Figure 3: Correction head architecture in the Kinetix environment. The MLP takes as input the current state, the base action, and a positional embedding indicating the index within the action chunk. It outputs a residual action that is added to the base action, yielding the refined action.

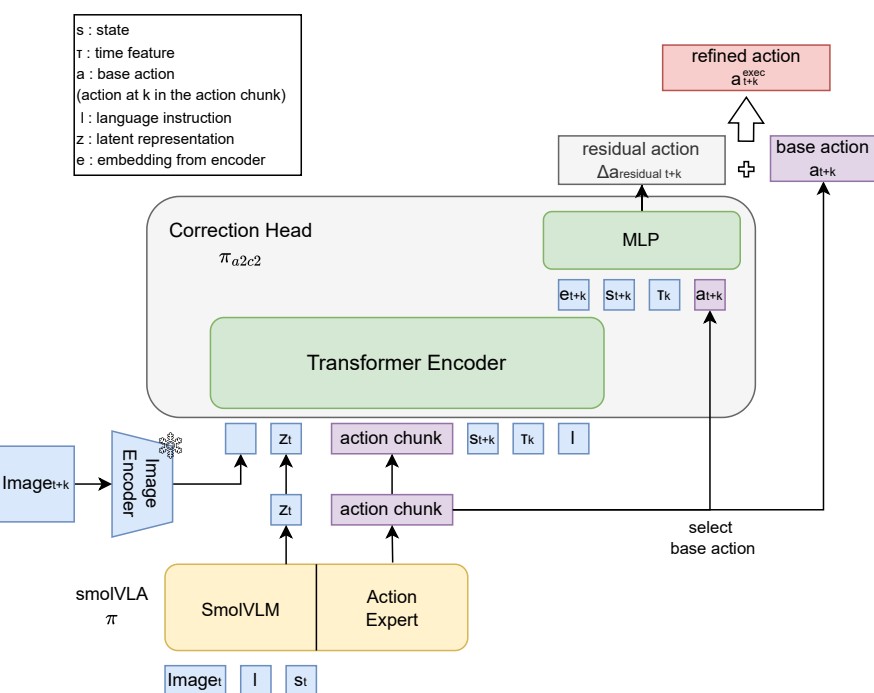

Figure 4: Architecture of the proposed Correction head in the LIBERO environment. The base policy is a SmolVLA that produces an action chunk from image, language instruction, and state inputs. A transformer encoder processes the latent representation from smolVLM $z_t$, state $s_{t+k}$, base action $a_{t+k}$, image feature, entire action chunk, time feature $\tau_k$, and language instruction $l$ to produce a latent representation $e_{t+k}$. A lightweight MLP predicts a residual action$\Delta a_{t+k}^{residual}$, based on the latent representation from the transformer encoder and base action $a_{t+k}$, state $s_{t+k}$, and time feature $\tau_k$. Then, added to the selected base action to obtain the refined action executed in the environment.

found within the LIBERO Spatial Benchmark. This task effectively measures the policy's ability to achieve robust grasping, spatial reasoning, and precise manipulation over a continuous action sequence. Please refer to the A.10 for further details.

## 4.2 MODEL TRAINING

In Kinetix, we used a flow-matching policy as the base model, following prior work on RTC Black et al. (2025). The Correction head network is a 3-layer multilayer perceptron (MLP). The input layer receives the concatenation of the state vector (2722-dim), the base action (6-dim), and the 2-dimensional sinusoidal positional feature. We did not use language instructions or latent representations from base policies, as the model was trained and evaluated separately for each task. Hidden layers have 512 units each with ReLU activation (Nair & Hinton, 2010) and layer normalization (Ba et al., 2016). The output layer produces a 6-dimensional residual vector, which is added element-wise to the base action. The total parameter count is 0.31M. Figure 3 shows the implementation detail for the Kinetix experiment.

For LIBERO spatial, we adopted SmolVLA (Shukor et al., 2025) (450M parameters) as the base, since it provides competitive performance among VLA models. The correction head consists of a transformer encoder and a lightweight MLP. Visual observations (top and wrist cameras) are encoded into 512-dimensional tokens using a ResNet-18 (He et al., 2016) pretrained on ImageNet (Deng et al., 2009). Language instructions are embedded by the smolVLM encoder provided in the base policy. The base action, latent features of the base policy, and the sinusoidal time embedding are also projected into 512-dim tokens. All tokens are concatenated and processed by a 6-layer transformer encoder. The pooled embedding, along with the base action and state vector, is passed through a 3-layer MLP (hidden size 512) to predict the residual action. The number of total parameters is 32M. Figure 4 shows the implementation detail for the LIBERO experiment. We also release the source code for both Kinetix and LIBERO experiments. See Appendix A.5 for the details.

Furthermore, to demonstrate that the effectiveness of our proposed method is not merely an artifact of addressing deficiencies in a weaker base model, we further extended our evaluation to a state-of-the-art VLA model. Specifically, we employed **Pi05** (Intelligence et al., 2025), utilizing the publicly available `pi05_libero_finetuned` checkpoint provided by LeRobot[1]. This model represents a significantly stronger baseline compared to SmolVLA, achieving high success rates on standard benchmarks.

## 5 RESULTS

### 5.1 KINETIX

We evaluate the proposed action chunk correction framework in the Kinetix benchmark under varying inference delays $d$ and execution horizons $e$. Figure 5 reports success rates aggregated across all 12 tasks. There are two baseline comparisons. First is Naive async. This strategy does not pay attention to the previous action chunk at all when generating a new one, naively switching chunks as soon as the new one is ready. Second is RTC.

As expected, both the naive async and RTC baselines degrade significantly as either the delay $d$ increases or the horizon $H$ becomes longer. In particular, when $d \geq 3$, the naïve baseline suffers a sharp drop in success rate due to compounding errors from executing outdated action chunks. RTC inference partially mitigates this issue by overlapping prediction and execution, but performance still declines as the execution horizon increases.

In contrast, the action chunk correction maintains consistently higher success rates across all settings. Because it refines each action using the most recent observation, the action chunk correction can counteract both the temporal misalignment introduced by inference delay and the drift that accumulates within long action horizons. For example, at delay $d = 4$, our proposed method achieves nearly 35% higher success than the naïve baseline, and remains above 85% even for horizons $H = 7$. This demonstrates that real-time correction of action chunks maintains performance both with inference delays and with long-horizon execution. Furthermore, we evaluated the integration of our proposed method with RTC. Since RTC and A2C2 are orthogonal in nature, they can be seamlessly combined to achieve both smooth chunk transitions and high-frequency reactivity. RTC primarily focuses on smoothing the predicted action chunks using guidance from previous chunks. In contrast, our correction head leverages the latest observations to ensure real-time reactivity. While both methods aim to mitigate the effects of inference latency, they address fundamentally different aspects of the problem. As demonstrated in Figure 5, this combination yields the highest overall success rate in the Kinetix simulation, significantly outperforming individual baselines.

### 5.2 LIBERO SPATIAL

Figure 6 and Table 3 summarize the evaluation on the LIBERO Spatial benchmark. We tested the Naïve async and A2C2 on this setting. Across 10 manipulation tasks with multimodal inputs, the correction head consistently improved success rates over the naïve baseline under both long horizons and injected delays. For example, with execution horizon $H = 40$ and delay $d = 10$, the naïve baseline achieved only 67% success, whereas the A2C2 reached 84%. Even when no delay was present, Action chunk correction provided notable gains at long horizons ($H = 50$, $d = 0$), raising success from 72.2% to 81.6%. These results demonstrate that residual refinement by correction head

---

[1]https://huggingface.co/lerobot/pi05_libero_finetuned

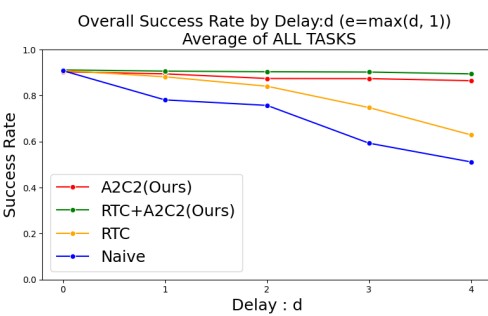 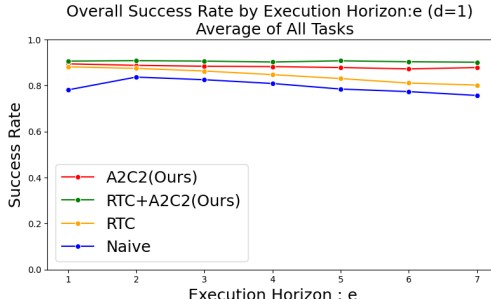

(a) Average Success rate as a function of inference delay $d$ with execution horizon fixed at $e = \max(d, 1)$. A2C2 (red) consistently outperforms both naive and real-time baselines, maintaining higher success rates even under large delays.

(b) Average Success rate as a function of execution horizon $e$ with delay fixed at $d = 1$. A2C2 (red) remains robust across horizons, while baselines degrade as horizon length increases.

Figure 5: Overall performance comparison in Kinetix tasks. Each data point averages over 2048 rollouts. Residual correction improves robustness under both increasing inference delay and longer execution horizons.

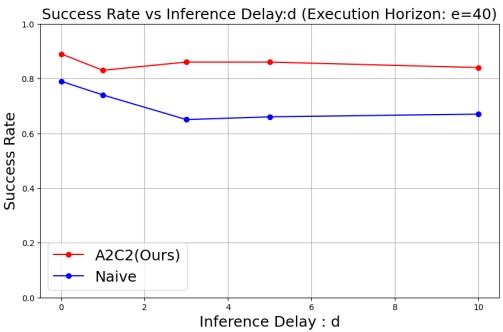 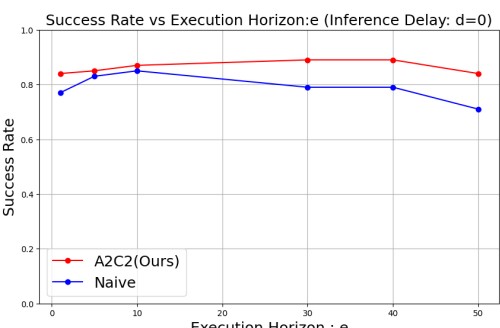

(a) Success Rate vs Inference Delay $d$ (Execution Horizon: e=40). A2C2 remains robust under inference delays.

(b) Success Rate vs Execution Horizon $e$ (Inference Delay: d=0). A2C2 consistently improves performance across horizons.

Figure 6: Results of LIBERO Spatial: Comparison of Success Rate under different conditions. (a) Effect of inference delay with fixed execution horizon. (b) Effect of execution horizon with no inference delay. Each data point is evaluated on 10 tasks, each with 10 rollouts, resulting in a total of 100 rollouts.

mitigates the degradation caused by outdated action chunks and restores closed-loop responsiveness, enabling large VLA models to maintain high success rates that require fine-grained spatial reasoning.

To investigate whether the efficacy of A2C2 generalizes to state-of-the-art foundation models, we further evaluated the method using the $\pi_{0.5}$ backbone. Table 2 summarizes the results on LIBERO Spatial with diffrent execution horizon $e$ and delay $d$.; The official success rate for LIBERO Spatial reported by Lerobot is $97.0\%$ and we also evaluated with execution horizon 5 and got $98.2\%$ success rate which shows state of the art result on LIBERO Benchmark. However, in the challenging scenario with a longer execution horizon ($e = 40$) and injected delay ($d = 10$), the baseline performance degrades to 82.2%. Crucially, A2C2 mitigates this drop, recovering the success rate to 86.4%. This result confirms that the benefits of action chunk correction are not limited to weaker policies; it provides robustness against execution latency even for high-performance SOTA models.

## 5.3 REAL WORLD TASK

We evaluate the proposed methods in the real world under long-horizon prediction. The task was to pick up the yellow cube and put it in the bowl. The result shows that the proposed methods

Table 1: LIBERO Spatial with smolVLA : success rate (%). 50 rollouts per task. Action chunk correction mitigates performance degradation under delay and long horizons.

| Method | Execution horizon $e$ | Delay $d$ | Success Rate (%) |
|---|---|---|---|
| Naïve | 10 | 0 | 81.8 |
| A2C2 (Ours) | 10 | 0 | **89.2** |
| Naïve | 40 | 10 | 64.4 |
| A2C2(Ours) | 40 | 10 | **84.2** |
| Naïve | 50 | 0 | 72.2 |
| A2C2(Ours) | 50 | 0 | **81.6** |

Table 2: LIBERO Spatial with $\pi_{0.5}$ : success rate (%). 50 rollouts per task. Action chunk correction mitigates performance degradation under delay and long horizons.

| Method | Execution horizon $e$ | Delay $d$ | Success Rate (%) |
|---|---|---|---|
| Naïve | 10 | 10 | 91.2 |
| A2C2 (Ours) | 10 | 10 | **92.0** |
| Naïve | 40 | 10 | 82.2 |
| A2C2(Ours) | 40 | 10 | **86.4** |
| Naïve | 50 | 0 | 90.6 |
| A2C2(Ours) | 50 | 0 | **92.4** |

outperform the naive baseline. By just adding the small correction head, we can utilize a long execution horizon.

## 6 RELATED WORK

**Imitation learning and VLAs:** Imitation learning (IL) trains agents from demonstrations provided by humans or expert policies, and has been a representative approach in learning robotic control (Osa et al., 2018). Recent advances have introduced generative sequence models to improve consistency and scalability. Diffusion Policy (Chi et al., 2023) utilizes diffusion models for action generation, enabling it to handle the multimodality of data distribution in imitation learning. In parallel, the Action Chunking Transformer (ACT) (Zhao et al., 2023) proposes a transformer-based policy that outputs action chunks rather than single-step actions, producing coherent behaviors while enabling faster inference. In addition, flow-based approaches, such as Flow Policy (Zhang et al., 2024), generate actions by learning continuous transport maps instead of iterative denoising.

Building on these foundations, a new class of vision–language–action (VLA) foundation models has emerged (Kawaharazuka et al., 2024), including $\pi_0$ Black et al. (2024), openVLA Kim et al. (2024), GR00T NVIDIA et al. (2025), and SmolVLA Shukor et al. (2025). These models adopt chunk-based prediction as the de facto standard for inference, similar to ACT (Zhao et al., 2023). Vision–Language–Action (VLA) models achieve broad task generalization by aligning multimodal inputs, but their architectures are considerably larger than diffusion- or transformer-based imitation policies. For instance, $\pi_0$ has about 3B parameters and openVLA around 7B, which makes inference latency significant even on modern GPU-accelerated hardware. While these models demonstrate the promise of scaling and multimodal grounding, their computational footprint exacerbates the latency problem in real-time control.

**Asynchronous chunk execution:** As model sizes increase, inference latency becomes a significant bottleneck, motivating asynchronous policy frameworks. In particular, the SmolVLA (Shukor et al., 2025) proposed a server–client architecture for mitigating inference delays. In this setup, the server receives observations and performs inference with a delay of $d$ control steps (including communication latency), then transmits an action chunk of horizon $H$ to the client. Then, the client executes these actions sequentially. However, because the $d$ delayed actions are not yet available at execution time, the client continues executing actions from the previous chunk until the new chunk arrives. This design ensures continuity but introduces the risk of inconsistency between consecutive chunks. For example, the earlier chunk may predict avoiding an obstacle by moving left, while the

Table 3: Real World Task: success rate (%). 50 rollouts. Action chunk correction mitigates performance degradation with long horizons.

| Method | Execution horizon $e$ | Success Rate (%) |
|---|---|---|
| Naïve | 20 | 38 |
| A2C2 (Ours) | 20 | **84** |
| Naïve | 40 | 26 |
| A2C2(Ours) | 40 | **58** |

newly received chunk may instead suggest moving right. Such mismatches across chunks can cause jerky motion and noticeable performance degradation, especially in dynamic environments.

To fix the chunk mismatches, Real Time Chunking (RTC) (Black et al., 2025) is proposed. It is an inference-time algorithm that enables smooth asynchronous execution for action-chunking policies by posing chunk switching as an inpainting problem. Specifically, it generates the next action chunk while executing the current one, "freezing" actions guaranteed to execute and "inpainting" the rest.

**Reducing inference latency:** One natural way to enhance a model's real-time performance is to reduce its inference time. Streaming Diffusion Policy (Høeg et al., 2024) or Streaming Flow Policy (Jiang et al., 2025) presents a new training procedure that enables faster inference. More generally, optimizations such as model compression (Lin et al., 2024) or memory optimization (Kwon et al., 2023) of models can also improve the inference speeds. However, as long as model scale and communication overhead prevent action generation from being faster than the control step, the challenges highlighted in this work remain unresolved.

## 7 CONCLUSION

In this paper, we propose Asynchronous Action Chunk Correction (A2C2), which introduces a lightweight action correction head by augmenting a large base policy, such as VLAs. A2C2 addresses the challenge of preserving reactivity under inference delays and long execution horizons of action chunking policies. The correction head is trained on the same dataset as the base policy and, in principle, can be added to any off-the-shelf VLAs. Our experiments in both the Kinetix simulation suite and the LIBERO Spatial benchmark demonstrated that Asynchronous Action Chunk Correction (A2C2) consistently maintained high success rates, even in settings where naïve or RTC degraded significantly.

While our approach adds minimal overhead compared to full model inference, further work is needed to validate its scalability to richer language instructions, out-of-distribution settings, and more dynamic tasks beyond those in LIBERO Spatial. Addressing these challenges would broaden the applicability of action chunk correction and strengthen its role as a general mechanism for enhancing reactivity in large policy architectures.

Recently, Large Language Models (LLMs) and Vision-Language Models (VLMs) have demonstrated improved generality through parameter scaling, as established by neural scaling laws (Kaplan et al., 2020). Since recent VLA policies are built upon these models, it is reasonable to expect that future VLAs will continue to grow in size to support deployment across diverse environments and tasks. Our work can be viewed as a step toward enabling such scaled VLAs to operate in real time without sacrificing responsiveness by introducing a lightweight correction mechanism that mitigates the effects of inference latency.

Moreover, inference of models with billions of parameters already exceeds the computational capacity of on-board processors on most robotic platforms. In practice, this motivates client–server architectures where the VLA runs on a remote server and the robot queries it over a network. In this setting, by explicitly treating communication delay as part of the inference latency in our formulation, our framework naturally extends to client–server architectures where large VLAs are executed remotely. Thus, our framework provides a pathway to leverage the generalization benefits of large-scale VLAs while still maintaining reactivity in real-world deployments, enabling the design of next-generation VLA systems that combine scalability with responsiveness.

ETHICS STATEMENT

This work does not involve human subjects, personally identifiable information, or sensitive data. All experiments were conducted on publicly available datasets.

REPRODUCIBILITY STATEMENT

We provide implementation details and dataset preprocessing in the Appendix, and full hyperparameter settings in the appendix. We released our source code for the experiments below:

- **Kinetix:** `https://anonymous.4open.science/r/a2c2-kinetix`
- **LIBERO:** `https://anonymous.4open.science/r/a2c2-smolvla`

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

# A APPENDIX

## A.1 KINETIX SIMULATION DETAIL

### A.1.1 ENVIRONMENT

We reused the 12 tasks from the Kinetix benchmark (Matthews et al., 2025) used in the RTC paper Black et al. (2025). A sample visualization of each of the environments is shown in Figure 7. The Kinetix environment has an observation space with 2722 dimensions, which does not include any images. Instead, it encodes information about polygons, circles, joints, thrusters, gravity, and the states of motors and thrusters described below. For entities not used in a given task, their corresponding entries are zero-padded. The action space has 6 dimensions. The first four correspond to motor controls, and the last two correspond to thruster controls. For unused actuators, their entries are set to zero via padding.

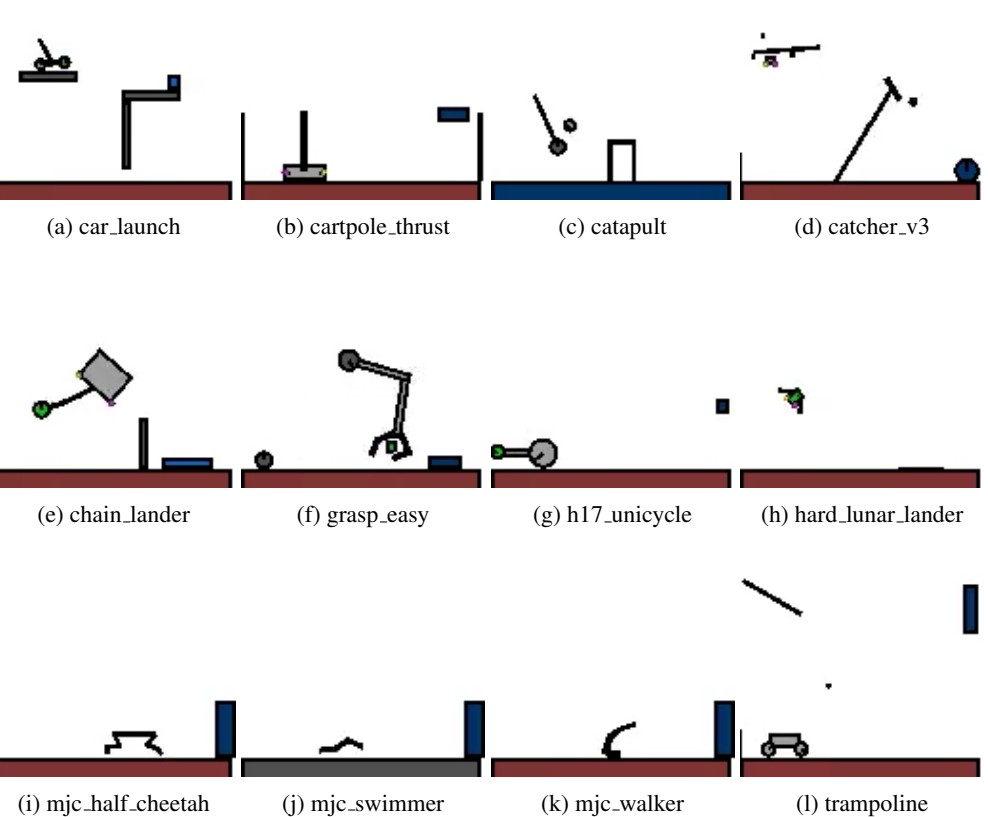

(a) car_launch      (b) cartpole_thrust      (c) catapult      (d) catcher_v3

(e) chain_lander      (f) grasp_easy      (g) h17_unicycle      (h) hard_lunar_lander

(i) mjc_half_cheetah      (j) mjc_swimmer      (k) mjc_walker      (l) trampoline

Figure 7: Visualization of the 12 tasks from the Kinetix simulation environment. Each subfigure corresponds to one task used in our experiments.

### A.1.2 DATASET GENERATION AND TRAINING DETAIL

An imitation learning dataset was required to test the flow policy and our correction head. In the Kinetix simulation, we follow the RTC implementation. First, we trained the expert policy with RPO (Rahman & Xue, 2022) on 8 seeds per task for 64 million environment steps each. For each task, we load the best-performing checkpoint for each seed and discard some seeds if they did not reach a certain success threshold. Then, we used the expert model to generate 1 million environment steps for each task. After that, we train the flow policy with the generated dataset. We saved the checkpoints for each, but used the last checkpoint for the evaluation.

Table 4: Training hyperparameters for the Kinetix flow policy.

| Hyperparameter | Value |
|---|---|
| Learning rate | $3 \times 10^{-4}$ |
| Gradient norm clip | 10.0 |
| Weight decay | $1 \times 10^{-2}$ |
| Warmup steps | 1000 |
| Batch size | 512 |
| Number of epochs | 32 |

Table 5: Training hyperparameters for the Kinetix Correction head.

| Hyperparameter | Value |
|---|---|
| Batch size | 512 |
| Number of epochs | 16 |
| learning rate | $1 \times 10^{-4}$ |
| weight decay | $1 \times 10^{-3}$ |
| Gradient norm clip | 5.0 |
| Warmup steps | 500 |

The correction head is then trained with the flow policy. The correction policy requires the base action from the base policy, so at every step, we infer the action chunk from the base policy and use it and the dataset to train the correction head. During the base flow policy training, we used a constant learning rate and added some warmup state. See Table 4 for more details on the settings. For the Correction Head training, we used the parameters shown in Table 5. In both the flow policy and A2C2 training, the AdamW optimizer (Loshchilov & Hutter, 2017) was used.

### A.1.3 EVALUATION DETAILS

In the evaluation, we rolled out 2048 per task and computed the success rate for different delays and execution horizon lengths. In the Kinetix simulation, we tested all combinations of delay and execution horizons compatible with the chosen action chunk size. All results are in Table 6.

### A.2 KINETIX EVALUATION WITH BIDIRECTIONAL DECODING AND TEMPORAL ENSEMBLE

While Real-Time Chunking (RTC) serves as our primary baseline, we also compare against other established action chunking strategies. Specifically, we evaluate Temporal Ensembling (TE) Zhao et al. (2023), which smoothes actions via a weighted average of predictions, and Bidirectional Decoding (BID) Liu et al. (2025), which enables closed-loop control with pre-trained policies via rejection sampling. To benchmark these methods, we conducted evaluations in the Kinetix simulation.

For the Bidirectional Decoding evaluation, we utilized a sample size of 16. As shown in Figure 8, Temporal Ensembling yields results that are comparable to or even inferior to the Naive baseline. Furthermore, while Bidirectional Decoding demonstrates a slight improvement over the Naive baseline, it consistently underperforms compared to both RTC and our proposed method. Notably, this shortfall occurs despite BID requiring significantly higher computational resources.

### A.3 KINETIX RESULT FOR EACH TASK

We mainly used the average success rate across tasks. To provide a more detailed analysis, Table 7 presents the breakdown of success rates for each individual task

Table 6: Kinetix: success rate (percent) under different execution horizons ($e$) and inference delays ($d$). 10 tasks and 10 rollouts per task. Residual correction consistently improves over the naïve baseline. The first, second, and third row of each cell denote the success rate of Naïve, RTC (Black et al., 2025), and A2C2(Ours), respectively.

| Delay ($d$) | Execution Horizon ($e$) | | | | | | | |
|---|---|---|---|---|---|---|---|---|
| | 1 | 2 | 3 | 4 | 5 | 6 | 7 | 8 |
| 0 | 90.8 | 90.4 | 89.8 | 88.9 | 88.1 | 87.4 | 86.6 | 86.0 |
| | 90.9 | 90.0 | 89.2 | 88.6 | 87.5 | 86.6 | 86.3 | 86.0 |
| | 90.3 | 89.5 | 89.6 | 88.9 | 88.8 | 88.9 | 88.2 | 87.8 |
| 1 | 78.1 | 83.7 | 82.6 | 80.9 | 78.5 | 77.4 | 75.7 | |
| | 88.2 | 87.5 | 86.3 | 84.8 | 83.1 | 81.1 | 80.2 | - |
| | 89.5 | 88.8 | 88.4 | 88.3 | 87.9 | 87.3 | 87.8 | |
| 2 | | 75.7 | 72.7 | 70.1 | 67.3 | 66.4 | | |
| | - | 84.1 | 81.4 | 79.3 | 76.4 | 74.8 | - | - |
| | | 87.4 | 87.5 | 87.5 | 87.1 | 86.6 | | |
| 3 | | | 59.3 | 59.5 | 56.5 | | | |
| | - | - | 74.8 | 71.0 | 67.5 | - | - | - |
| | | | 87.4 | 87.2 | 86.0 | | | |
| 4 | | | | 51.2 | | | | |
| | - | - | - | 62.9 | - | - | - | - |
| | | | | 86.5 | | | | |

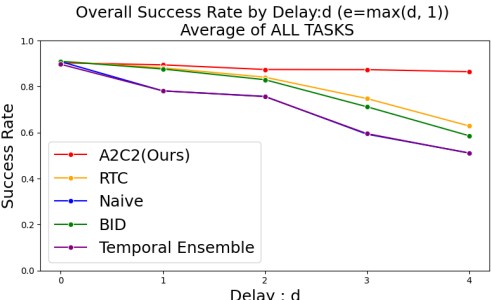

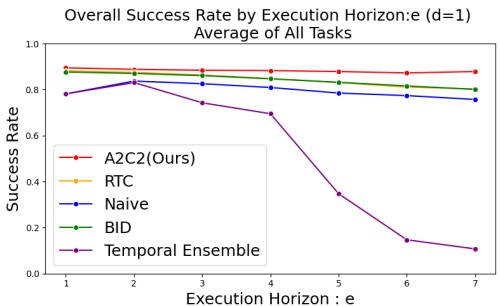

(a) Average Success rate as a function of inference delay $d$ with execution horizon fixed at $e = \max(d, 1)$.

(b) Average Success rate as a function of execution horizon $e$ with delay fixed at $d = 1$.

Figure 8: Overall performance comparison in Kinetix tasks. Each data point averages over 2048 rollouts.

## A.4 LIBERO SIMULATION DETAIL

### A.4.1 ENVIRONMENT

LIBERO Spatial consists of 10 tasks. We evaluated all tasks, and the corresponding language instructions are listed below. The language instructions are:

1. pick up the black bowl between the plate and the ramekin and place it on the plate
2. pick up the black bowl next to the ramekin and place it on the plate
3. pick up the black bowl from the table center and place it on the plate
4. pick up the black bowl on the cookie box and place it on the plate
5. pick up the black bowl in the top drawer of the wooden cabinet and place it on the plate
6. pick up the black bowl on the ramekin and place it on the plate

Table 7: Performance comparison across varying Kinetix tasks. The average success rate of 2048 rollouts. Our proposed method (Residual) consistently outperforms baselines including Naive, Real-Time Chunking (RTC), Bidirectional Decoding (BID), and Temporal Ensembling (TE). This result is under inference delay=4 and execution horizon=4.

| Task | Naive | RTC | BID | TE | Residual (Ours) |
|---|---|---|---|---|---|
| Cartpole Thrust | 0.292 | 0.388 | 0.323 | 0.292 | **0.999** |
| Grasp Easy | 0.837 | 0.874 | 0.881 | 0.841 | **0.979** |
| Hard Lunar Lander | 0.320 | 0.475 | 0.440 | 0.325 | **0.727** |
| Car Launch | 0.538 | 0.584 | 0.565 | 0.540 | **0.998** |
| Half Cheetah | 0.680 | 0.822 | 0.805 | 0.658 | **0.937** |
| Chain Lander | 0.914 | 0.932 | 0.918 | 0.917 | **0.982** |
| Walker | 0.054 | 0.365 | 0.274 | 0.054 | **0.484** |
| Catcher V3 | 0.259 | 0.471 | 0.334 | 0.259 | **0.924** |
| Swimmer | 0.282 | 0.537 | 0.434 | 0.282 | **0.929** |
| Catapult | 0.292 | 0.361 | 0.325 | 0.292 | **0.474** |
| Trampoline | 0.841 | 0.887 | 0.863 | 0.843 | **0.950** |
| H17 Unicycle | 0.831 | 0.854 | 0.874 | 0.827 | **0.996** |

7. pick up the black bowl next to the cookie box and place it on the plate

8. pick up the black bowl on the stove and place it on the plate

9. pick up the black bowl next to the plate and place it on the plate

10. pick up the black bowl on the wooden cabinet and place it on the plate

### A.4.2 DATASET AND TRAINING DETAIL

We used the LIBERO Dataset with the LeRobot dataset format available on Huggingface and we used the LeRobot framework to read the dataset. LeRobot also has a well-organized training pipeline and makes it easy to create and try new architectures.

First, we trained SmolVLA as a base policy. There is an option for training the policy from scratch or fine-tuning the pretrained model. In our setting, we chose the training from scratch because SmolVLA is pretrained mainly with S0-101, which is a different embodiment from the Franka arm used in the LIBERO benchmark.

In the Kinetix simulation, the base policy predicts the action chunk every time in the correction head training. However, it is too time-consuming with a large VLA model. Then, we added the inference result of SmolVLA on the dataset for training the correction head. The new dataset has all the LIBERO Spatial data, the action chunk result, and the VLM latent representation from the SmolVLA policy for each step.

After that, we trained the correction head with the dataset we created. For SmolVLA training, we trained a model from scratch with a cosine learning scheduler, which is the default setting for SmolVLA training. The parameter for SmolVLA training is in Table 8

For Correction head training, we use a constant learning rate of 1e-5. High learning rates, such as 1e-4, do not work well for the Correction head training. See Table 9

In both SmolVLA and Correction head training, the AdamW optimizer was used (Loshchilov & Hutter, 2017).

### A.4.3 EVALUATION DETAIL

For the evaluation, we tested various combinations of delay steps and horizon steps first. We tested 10 rollouts per task, and LIBERO Spatial has 10 tasks. Then, to evaluate more precisely, we select 3 pairs of delay and horizon, (0,10), (10,40), (0,50), and rollouts 50 per task. All results for LIBERO Spatial are shown in Table 10.

Table 8: Training hyperparameters for LIBERO with SmolVLA.

| Hyperparameter | Value |
| --- | --- |
| Learning rate | $1 \times 10^{-4}$ |
| Scheduler | Cosine |
| Warmup steps | 1000 |
| Decay steps | 30000 |
| Minimum learning rate | $2.5 \times 10^{-6}$ |
| Batch size | 64 |
| Training steps | 100000 |
| Optimizer $\epsilon$ | $1 \times 10^{-8}$ |
| Optimizer weight decay | $1 \times 10^{-10}$ |
| Gradient norm clip | 10 |

Table 9: Training hyperparameters for LIBERO Correction head.

| Hyperparameter | Value |
| --- | --- |
| Learning rate | $1 \times 10^{-5}$ (constant) |
| Batch size | 64 |
| Training steps | 200000 |
| Optimizer weight decay | $1 \times 10^{-5}$ |
| Model dimension | 512 |
| Number of heads | 8 |
| Number of encoder layers | 6 |

## A.5 SOURCE CODE FOR EXPERIMENTS

To facilitate reproducibility, we have released the source code for our experiments:

- **Kinetix:** https://anonymous.4open.science/r/a2c2-kinetix
- **LIBERO:** https://anonymous.4open.science/r/a2c2-smolvla

## A.6 SIMPLE MLP ARCHITECTURE FOR LIBERO SPATIAL

We employed distinct architectures for the two benchmarks to address their specific characteristics. While Kinetix is a single-task, state-based simulation, LIBERO Spatial involves multimodal inputs, including vision and language. Although we initially used a Transformer for LIBERO Spatial to handle this rich information, we explored a streamlined MLP architecture to facilitate real-time inference. This MLP has 0.36M parameters, comparable to the model used for Kinetix. In this design, each input modality (images, state, base action, time, and VLM features) is projected via a linear layer to a common dimension. The fused vector is then processed by a shallow MLP to predict the residual action. Then, we trained this model with a learning rate of 1e-5 for 200k steps. After training, we evaluated the model with the LIBERO Spatial benchmark.

Figure 9 demonstrates that the simple MLP architecture fails to maintain the base model's performance and yields significantly inferior results compared to our proposed architecture. This suggests that a shallow MLP lacks the sufficient representational capacity to capture the complex multimodal dependencies required for the LIBERO Spatial task. While our proposed architecture is significantly smaller than the base model and yet successfully retains its performance, the Simple MLP fails to achieve this balance. This indicates that the Simple MLP is overly simplified and lacks the necessary representational capacity to handle the complex multimodal inputs of the LIBERO benchmark.

## A.7 RTC ON LIBERO SPATIAL

We evaluated the performance of Real Time Chunking (RTC) on the LIBERO Spatial benchmark using SmolVLA, the same model employed in our main experiments. To conduct these experiments,

Table 10: LIBERO Spatial: success rate under different execution horizons and inference delays. 10 tasks and 10 rollouts per task. Residual correction consistently improves over the naïve baseline.

| Execution Horizon | Inference Delay $d$ | Naïve | A2C2 (Ours) |
|---|---|---|---|
| 40 | 10 | 0.67 | 0.84 |
| 40 | 5 | 0.66 | 0.86 |
| 40 | 3 | 0.65 | 0.86 |
| 40 | 1 | 0.74 | 0.83 |
| 10 | 10 | 0.75 | 0.88 |
| 10 | 5 | 0.82 | 0.92 |
| 10 | 3 | 0.81 | 0.89 |
| 10 | 1 | 0.83 | 0.92 |
| 50 | 0 | 0.71 | 0.84 |
| 40 | 0 | 0.79 | 0.89 |
| 30 | 0 | 0.79 | 0.89 |
| 10 | 0 | 0.85 | 0.87 |
| 5 | 0 | 0.83 | 0.85 |
| 1 | 0 | 0.77 | 0.84 |

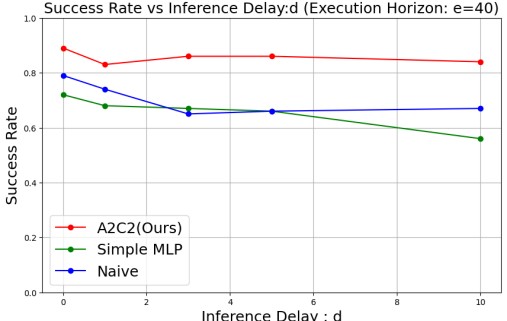 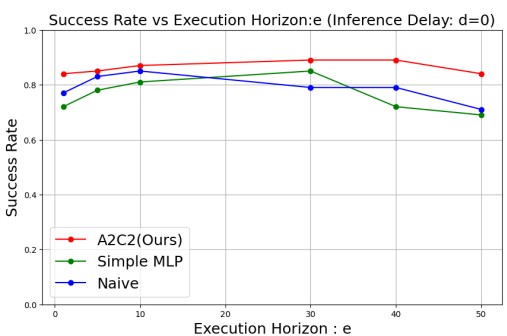

(a) Success Rate vs Inference Delay $d$ (Execution Horizon: e=40). A2C2 remains robust under inference delays, but the simple MLP architecture couldn't maintain its performance.

(b) Success Rate vs Execution Horizon $e$ (Inference Delay: d=0). A2C2 consistently improves performance across horizons, but the Simple MLP architecture couldn't maintain its performance.

Figure 9: Results of LIBERO Spatial: Comparison of Success Rate under different conditions. (a) Effect of inference delay with fixed execution horizon. (b) Effect of execution horizon with no inference delay. Each data point is evaluated on 10 tasks, each with 10 rollouts, resulting in a total of 100 rollouts.

we updated our codebase to the latest version of LeRobot, which includes an RTC implementation for diffusion and flow-based policies. Using the same experimental setup, we evaluated SmolVLA + RTC with 50 rollouts per task. Regarding hyperparameters, we followed the default RTC settings, utilizing a maximum guidance weight of 10 and an exponential prefix attention schedule.

Although RTC provides marginal gains under latency, it fails to fundamentally resolve the issue where actions are grounded in outdated observations caused by long horizons and delays. As a result, it still suffers from performance degradation. Our method, however, effectively corrects for this temporal misalignment, demonstrating robust performance.

## A.8 COMPUTATIONAL RESOURCES

We trained both models on NVIDIA RTX A6000 and H200 GPUs. Training in Kinetix required about 20 minutes per task on A6000, while LIBERO residual training (200k steps) took about 4 hours on H200.

Table 11: LIBERO Spatial: success rate (%). 50 rollouts per task. Action chunk correction mitigates performance degradation under delay and long horizons.

| Method | Execution horizon $e$ | Delay $d$ | Success Rate (%) |
|---|---|---|---|
| Naïve | 10 | 0 | 81.8 |
| RTC | 10 | 0 | 81.2 |
| A2C2 (Ours) | 10 | 0 | **89.2** |
| Naïve | 40 | 10 | 64.4 |
| RTC | 40 | 10 | 66.0 |
| A2C2(Ours) | 40 | 10 | **84.2** |

## A.9 INFERENCE TIME COMPARISON

We benchmarked the average inference time per step for the base policies—SmolVLA (450M parameters) and $\pi_{0.5}$ (using the $\pi_{0.5}$ checkpoint)—against our Correction Head (32M parameters). Measurements were conducted over 100 trials each on an NVIDIA RTX 5080 laptop GPU (16GB VRAM). As shown in Table 12, our correction head operates with negligible latency (4.7 ms) com-

Table 12: Average inference time per step (seconds). Computed over 100 trials.

| Model | Avg. Inference Time |
|---|---|
| SmolVLA (Base Policy) | 101 msec |
| $\pi_{0.5}$ (Base Policy) | 197 msec |
| Correction Head (**Ours**) | **4.7 msec** |

pared to the base policies. Notably, while the stronger $\pi_{0.5}$ baseline requires 197 ms per step—nearly double the time of SmolVLA (101 ms)—our method maintains a $\sim 40\times$ speed advantage over $\pi_{0.5}$. This demonstrates that our approach can be seamlessly integrated into high-frequency control loops (e.g., $> 100$ Hz) even when paired with computationally heavier SOTA foundation models, effectively bridging the gap between slow, high-level reasoning and fast, real-time actuation.

## A.10 REAL WORLD EXPERIMENT DETAIL

### A.10.1 HARDWARE SETUP

We use the SO-101 arm for the real-world evaluation. The selected task is a simple pick-and-place task. We use the 3-color, red, blue, and yellow cube as an object. For the visual information, we use the RealSense camera for the top, and use usb camera for the wrist camera. Figure 10 illustrates the hardware setup.

### A.10.2 DATA COLLECTION

We follow the leader-follower style of teleoperation for the data collection. For the pick and place task, we selected the task of picking up the yellow block and placing it into the bowl. We use the 'pick up the yellow cube and put in the bowl' for the language instruction. For this single task, we collected 100 episodes. To ensure generalization, the cubes and the bowl were repositioned randomly in every episode.

### A.10.3 TRAINING

For the training, we used the 'lerobot/smolvla_base' as a base model and finetuned it with 40k steps. Please see Table 13 for the smolVLA training details. Also, for the correction head training, we use the same training parameter as Table 9.

### A.10.4 DEPLOYMENT

Experiments were conducted using varying prediction horizon lengths during deployment. Furthermore, while object locations were randomized during the data collection phase, we set the cubes

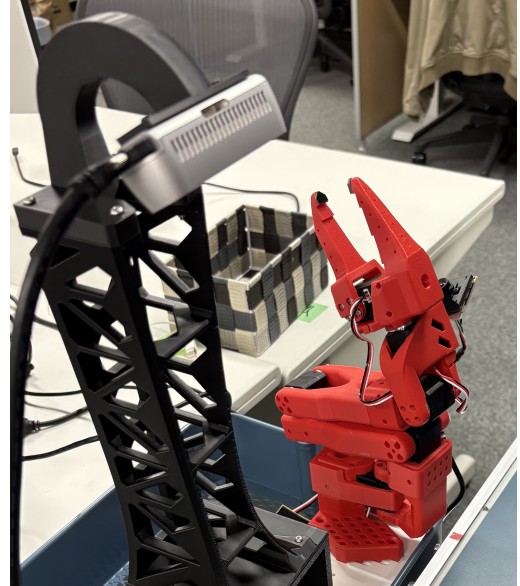
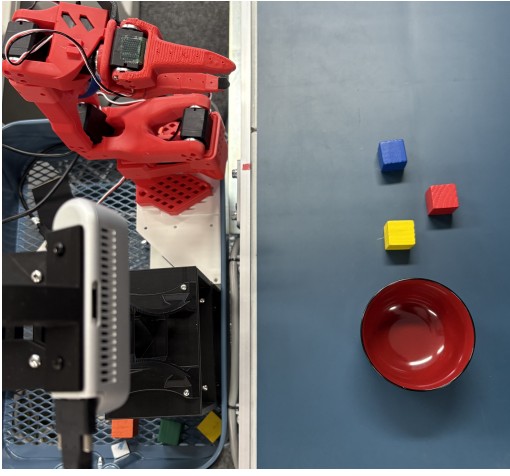

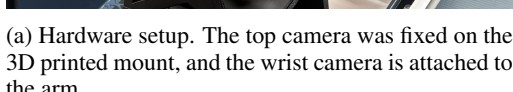

(a) Hardware setup. The top camera was fixed on the 3D printed mount, and the wrist camera is attached to the arm.

(b) Object for the pick and place task. This position is used for the evaluation.

Figure 10: Hardware setup for the real-world evaluation. To fix the robot and camera position, we installed the arm and top camera on the IKEA cart. The small colorful cubes and bowl were selected for the pick and place task.

Table 13: Training hyperparameters for real-world experiment with SmolVLA.

| Hyperparameter | Value |
|---|---|
| Learning rate | $1 \times 10^{-4}$ |
| Scheduler | Cosine |
| Warmup steps | 1000 |
| Decay steps | 30000 |
| Minimum learning rate | $2.5 \times 10^{-6}$ |
| Batch size | 64 |
| Training steps | 40000 |
| Optimizer $\epsilon$ | $1 \times 10^{-8}$ |
| Optimizer weight decay | $1 \times 10^{-10}$ |
| Gradient norm clip | 10 |

and the bowl to a fixed position (as illustrated in Figure 10b) for the evaluation phase to ensure fair comparison across different conditions. Under this fixed setup, we collected 50 episodes for each experimental condition. Success was defined as the robot correctly grasping the block and dropping it into the bowl, and the performance was evaluated based on the success rate.

### A.10.5 QUALITATIVE ANALYSIS FOR REALWORLD

We present a qualitative comparison in a real-world pick-and-place task with a long execution horizon ($H = 40$). As illustrated in Figure 11a, the Naive implementation suffers from compounding errors, causing the robot to drift and eventually miss the grasp. In contrast, Figure 11b demonstrates the robustness of A2C2. The residual correction mechanism effectively compensates for trajectory errors in real-time without over-correcting or introducing instability, enabling the robot to precisely grasp the target object despite the extended open-loop duration.

(a) Sequential frames of Naive implementation with 40 execution horizon. It fails to grasp the cube.

(b) Sequential frames of A2C2 implementation with 40 execution horizon. The correction head adjusts its action and successfully grasps the cube.

Figure 11: Sequential frames for Real World Evaluation.

## A.11 THE USE OF LARGE LANGUAGE MODELS

We used Large Language Models to polish our writing.

