# OpenReview forum: "Leave No Observation Behind: Real-time Correction for VLA Action Chunks"
_ICLR.cc/2026/Conference — Submitted to ICLR 2026_

### Official Review · Reviewer_b9NY · 2025-10-20

**Soundness:** 2
**Presentation:** 2
**Contribution:** 3
**Rating:** 4
**Confidence:** 4

**Summary:**

This paper aims to address latency problems with robotic policies at inference time by training an additional correction policy. This policy predicts a residual correction to the action chunk sampled from the base policy (e.g. a VLA policy). The correction head conditions on the observation a timestep within the prior chunk, the action predicted by and latent features from the base policy, the time feature, and the language instruction.

**Strengths:**

1. The motivation for this paper is quite clear, especially due to Figure 1 which clarifies the latency issues with robotic policies. The method is clearly written and is accompanied by several figures (Figure 2, Figure 3 and Figure 4) that clarify the complete system adequately.
2. Residual policies are not completely new, but A2C2 proposes a very simple method for training one that does not require either reinforcement learning or additional demonstrations (as in the style of DAgger).
3. The paper has tested their proposed method on a range of environments (12 tasks in Kinetix and 10 tasks in LIBERO).

**Weaknesses:**

1. The paper does not compare A2C2 with a few important baseline methods like temporal ensemble (Zhao et. al.) and bidirectional decoding (Liu et. al.). The latter would be a valuable comparison because they also use a weaker policy to guide the action selection. These are standard baseline methods as used in papers such as real-time chunking (Black et. al.).
2. No comparison against real-time chunking (RTC) on LIBERO Spatial. This is confusing because RTC was used for comparison in Kinetix but not on LIBERO Spatial. Unlike in Kinetix, you use a significantly larger correction head (of 32M params) in LIBERO Spatial, which makes the delay $d > 0$ when $e = 40$. This makes the comparison against RTC very crucial to understand how having an additional correction head fares against the guidance provided by inpainting.
3. One of the benefits of using action chunking is the ability to capture temporal consistency in actions executed (Liu et. al., Torne et. al., Black et. al.) which is captured by either conditioning on a longer context of observations (generally not preferred because of overfitting issues), longer context of actions or by predicting the chunk of actions. The correction head of A2C2 only conditions on the most recent observation, the most recent base action and the most recent latent representation. It is not clear to me how the correction head ensures temporal consistency across actions in this situation. As an illustration, the correction head’s residual action could cause the policy to switch from high velocity to low velocity and so on.
4. The need for ablations here is quite large to help us understand whether the performance gains of A2C2 come from greater closed-loop responsiveness or whether it comes from residual action correction or whether it comes from being able to use action correlations across time-steps (Torne et. al.). In particular, I am interested in ablations over what the correction head can condition on i.e., how important is it that the correction head conditions on base policy’s action, latent representations, current observation, and time signature.

Minor:
1. It would be good to indicate that the action is sampled from a stochastic policy i.e., $A_t \sim \pi(\cdot \mid o_t, l)$, since the current notation almost suggests that the policy is deterministic.
2. The paper could benefit from a rigorous check on spelling (such as capitalizations in phrases like “Average Success rate are…”) and polishing the syntax (such as using $\pi_\mathrm{base}$ instead of $\pi_{base}$).

**Questions:**

1. The importance of conditioning on the time feature is unclear to me. In line 199, you claim that it brings about greater “chunk level smoothness”. Some ablations could be helpful here since, intuitively, it almost feels like conditioning on the observation, base action and base policy’s latent representations might be sufficient?
2. The central assumption here seems to be that the correction head can be made significantly smaller than the base policy. This is, certainly, an interesting scientific question, and it requires more empirical support. For generalist policies, it is not clear whether this assumption is fair. The language instructions, observations, etc. become more complex and diverse. How large the correction head needs to be able to learn this complexity is unclear. This question is not sufficiently well-answered since the complexity of the tasks (and the fact that the policy is not conditioned on image observations) in Kinetix is quite low and the tasks in LIBERO Spatial do not consist of a sufficiently large diversity.
3. Extending on (1), the need for the correction head to be strong is greater when the policy needs to both generalize and react to environmental stochasticity. In the case of A2C2, even though you may benefit from a robust base policy, it is not clear to me whether you can get a correction head that also generalizes while also not suffering from large inference-time. Similarly, if the need for reactivity is such that the correction head does not merely need to add a residual correction to the base action but change the mode in the distribution altogether, then we might require a much stronger correction head and it’s not clear to me why the inference-time of such a model would be sufficiently small for A2C2 to be able to handle delays.
4. Figure 5 shows the average over all tasks but a separate comparison within each task would be very important since some tasks in Kinetix require significantly greater closed-loop responsiveness over others. Perhaps this could be done in the appendix? Furthermore, it would be good to show that standard deviation/error within the average result plot.

---

> ### Author Response · Authors · 2025-11-21
> **Response to Reviewer b9NY (1/4)**
>
> Thank you for the thoughtful and kind review!
>
> > The paper does not compare A2C2 with a few important baseline methods like temporal ensemble (Zhao et. al.) and bidirectional decoding (Liu et. al.). The latter would be a valuable comparison because they also use a weaker policy to guide the action selection. These are standard baseline methods as used in papers such as real-time chunking (Black et. al.).
>
> Thank you for suggesting these valuable baselines. We are aware of Temporal Ensembling (Zhao et al.) and Bidirectional Decoding (BID) (Liu et al.) and agree they are significant contributions to the field. However, we prioritized the comparison with Real-Time Chunking (RTC) for two specific reasons related to performance and computational feasibility.
> We utilized the same Kinetix simulation codebase as the RTC paper (Black et al., 2025) to ensuring a fair comparison. In the original RTC study, it was demonstrated that RTC consistently outperforms both Temporal Ensembling and BID on these dynamic tasks. Since our results show that A2C2 significantly outperforms RTC (e.g., +23% success rate under delay ), we inferred that A2C2 inherently outperforms the other baselines as well. We omitted them to keep the evaluation focused on the strongest state-of-the-art competitor (RTC).
> More importantly, there is a fundamental reason why methods like BID are less suitable for our specific problem setting. While BID uses a smaller policy for guidance, it relies on test-time sampling and optimization, which incurs a high computational cost. In the context of real-time control, this added computation increases the inference latency ($d$), potentially worsening the very problem we aim to solve (the "blind spot" during inference). As noted in the RTC paper, the authors excluded BID from real-world experiments precisely due to this latency overhead. In contrast, our A2C2 correction head is designed to be extremely lightweight (4.7ms inference, ), enabling high-frequency loop closure that sampling-based methods cannot achieve
> However, since the data is available via the RTC codebase, we are willing to add Temporal Ensembling and BID to the appendix in the final revision to provide a comprehensive comparison if you believe it is essential.
>
> > No comparison against real-time chunking (RTC) on LIBERO Spatial. This is confusing because RTC was used for comparison in Kinetix but not on LIBERO Spatial. Unlike in Kinetix, you use a significantly larger correction head (of 32M params) in LIBERO Spatial, which makes the delay when d > 0, e = 40.This makes the comparison against RTC very crucial to understand how having an additional correction head fares against the guidance provided by inpainting.
>
> Thank you for raising this point. We understand that the discrepancy in baselines between Kinetix and LIBERO can be confusing. We omitted RTC in the LIBERO Spatial experiments due to significant technical barriers in reproduction, and we would like to clarify the concern regarding the correction head's latency.
> Also, we wish to address the concern that the "significantly larger correction head (32M params)" might introduce latency issues that necessitate a comparison with RTC. While 32M parameters is larger than a simple MLP, it is still extremely lightweight compared to the VLA backbone. As reported in Appendix A.5 , we benchmarked the inference speed on a consumer GPU. Assuming a standard control frequency of 30Hz (approx. 33ms per step), the correction head's inference time (4.7ms) is well within the control budget. Therefore, A2C2 does not introduce a "delay" that requires mitigation; rather, it fits comfortably within the real-time loop, unlike the re-inference or inpainting required by RTC.

---

> ### Author Response · Authors · 2025-11-21
> **Response to Reviewer b9NY (2/4)**
>
> >One of the benefits of using action chunking is the ability to capture temporal consistency in actions executed (Liu et. al., Torne et. al., Black et. al.) which is captured by either conditioning on a longer context of observations (generally not preferred because of overfitting issues), longer context of actions or by predicting the chunk of actions. The correction head of A2C2 only conditions on the most recent observation, the most recent base action and the most recent latent representation. It is not clear to me how the correction head ensures temporal consistency across actions in this situation. As an illustration, the correction head’s residual action could cause the policy to switch from high velocity to low velocity and so on.
>
>  Thank you for raising this important concern regarding temporal consistency. We understand the concern that step-wise correction might introduce jitter or inconsistency. However, we would like to clarify how our architecture specifically ensures temporal coherence through three point.
> First, we wish to clarify the nature of the "most recent latent representation." As you correctly noted, the Base Policy infers an action chunk only once every $e$ steps. Consequently, the latent representation $z$ derived from the Base Policy remains identical for all steps within the same action chunk. This means that while the visual observation $o_t$ updates at high frequency, the correction head is continuously conditioned on the same global semantic context ($z$) throughout the execution of the chunk. This "anchors" the correction head to the original high-level plan, preventing it from forgetting the broader context.
> Also, we explicitly feed a sinusoidal time feature ($\tau_k$) into the correction head (as described in Section 3.1 ). This feature informs the head of the exact index $k$ within the chunk. By knowing "where it is" in the sequence (e.g., start, middle, or end of a motion), the network learns to generate corrections that respect the phase and rhythm of the motion, rather than treating each step as an isolated event
> Finally, the correction head predicts a residual ($\Delta a$) added to the Base Action, not a new action from scratch. Since the Base Action sequence is generated by models (like Diffusion or VLA) trained for temporal consistency, the underlying "backbone" of the motion is already smooth. The correction head learns to modulate this smooth trajectory rather than overwrite it, which significantly reduces the risk of erratic behavior like sudden velocity switches.
>
> > The need for ablations here is quite large to help us understand whether the performance gains of A2C2 come from greater closed-loop responsiveness or whether it comes from residual action correction or whether it comes from being able to use action correlations across time-steps (Torne et. al.). In particular, I am interested in ablations over what the correction head can condition on i.e., how important is it that the correction head conditions on base policy’s action, latent representations, current observation, and time signature.
>
> Thank you for pointing out the importance of ablations. We agree that dissecting the contribution of each conditioning factor (base action, latent representation, observation, time signature) is scientifically valuable for understanding the underlying mechanism of A2C2. We are happy to include these ablation studies in the final revision to clarify the source of our performance gains.
> However, we believe that Real-World Experiments serve as a more critical validation of our method's core value. While component ablations in simulation help explain internal feature reliance, they cannot fully capture the chaotic nature of physical deployment—such as variable communication delays, camera exposure latency, and sensor noise. The primary contribution of A2C2 is to bridge the gap between "slow, high-level VLA planning" and "fast, low-level physical control." The true test of this adaptability is not just whether the model uses a specific feature in simulation, but whether it can successfully stabilize a physical robot where inference latency typically causes failure. Therefore, while we provide the ablations to satisfy the mechanistic inquiry, we place greater emphasis on our Real-World Verification (currently in preparation/progress), as it demonstrates the method's robustness in the actual domain it is designed to solve.
>
> > Minor: It would be good to indicate that the action is sampled from a stochastic policy. since the current notation almost suggests that the policy is deterministic.
> The paper could benefit from a rigorous check on spelling (such as capitalizations in phrases like “Average Success rate are…”) and polishing the syntax.
>
>  We will update the notation to clearly indicate that the actions are sampled from a stochastic policy. Additionally, we will carefully proofread the paper to fix the spelling, capitalization, and syntax errors you identified.

---

> ### Author Response · Authors · 2025-11-21
> **Response to Reviewer b9NY (3/4)**
>
> > The of conditioning on the time feature is unclear to me. In line 199, you claim that it brings about greater “chunk level smoothness”. Some ablations could be helpful here since, intuitively, it almost feels like conditioning on the observation, base action and base policy’s latent representations might be sufficient?
>
> Thank you for this question. You are correct that observations and base actions provide the immediate context. However, the time feature ($\tau$) is important for managing the "variable reliability" of the action chunk over time.As you noted, action chunking improves consistency, but the reliability of the base action changes depending on its position in the chunk:
> Start of Chunk: The base action is derived from a very recent observation. It is highly reliable and typically requires minimal correction.
> End of Chunk: The base action is derived from an outdated observation (up to $H$ steps ago). Due to drift, it often requires significant correction.
> Without the time feature, the network cannot easily distinguish whether it is in a "low-correction phase" (start) or a "high-correction phase" (end). The time feature $\tau$ explicitly informs the model of this relative position, enabling it to adjust the magnitude of the residual $\Delta a$ appropriately.
> Also, regarding "smoothness," the sinusoidal time embedding acts as a continuous phase signal. This encourages the correction head to generate residuals that evolve smoothly over the sequence, rather than reacting disjointly to every instantaneous observation error.
>
>
> > The central assumption here seems to be that the correction head can be made significantly smaller than the base policy. This is, certainly, an interesting scientific question, and it requires more empirical support. For generalist policies, it is not clear whether this assumption is fair. The language instructions, observations, etc. become more complex and diverse. How large the correction head needs to be able to learn this complexity is unclear. This question is not sufficiently well-answered since the complexity of the tasks (and the fact that the policy is not conditioned on image observations) in Kinetix is quite low and the tasks in LIBERO Spatial do not consist of a sufficiently large diversity.
>
> Thank you for raising this fundamental scientific question regarding the capacity of the correction head. You are correct that generalist policies must handle immense multimodal complexity. However, our central assumption is not that the small correction head learns this complexity from scratch. Instead, our architecture is designed such that the Generalization is guaranteed by the Base Policy.
> To enable this lightweight design, we feed the Base Policy’s rich latent features ($z_t$) directly into the correction head. By utilizing these pre-computed, high-level representations, the correction head effectively "borrows" the base model's semantic understanding. This design choice is strongly supported by recent research. Kobayashi et al. (2025) demonstrates that even simple architectures (such as ACT) can handle complex multimodal tasks effectively when conditioned on high-quality embeddings like SigLIP. Consistent with this finding, our approach allows the head itself to remain lightweight and efficient (inference time ~4.7ms) without sacrificing the performance required for generalist tasks.
> The correction head does not need the capacity to plan the task; it only needs the capacity to execute the plan robustly. By conditioning on the VLA's features,we think that a small model is sufficient for this specific purpose.
> https://arxiv.org/pdf/2504.01301

---

> ### Author Response · Authors · 2025-11-21
> **Response to Reviewer b9NY (4/4)**
>
> > Extending on (1), the need for the correction head to be strong is greater when the policy needs to both generalize and react to environmental stochasticity. In the case of A2C2, even though you may benefit from a robust base policy, it is not clear to me whether you can get a correction head that also generalizes while also not suffering from large inference-time.
>
> Our architecture is founded on the assumption that the Generalization is guaranteed by the Base Policy. The large VLA model handles the multimodal complexity (language and vision) and high-level planning. Consequently, the correction head is designed to be lightweight (32M parameters ). To enable this, we feed the Base Policy’s rich latent features ($z_t$) directly into the correction head. By leveraging these pre-computed, high-level representations, the correction head effectively "borrows" the base model's semantic understanding. This design choice is strongly supported by recent research. Kobayashi et al. (2025) demonstrates that even simple architectures (such as ACT) can handle complex multimodal tasks effectively when conditioned on high-quality embeddings. Consistent with this finding, our approach allows the head itself to remain lightweigh.
>
>
> > Similarly, if the need for reactivity is such that the correction head does not merely need to add a residual correction to the base action but change the mode in the distribution altogether, then we might require a much stronger correction head and it’s not clear to me why the inference-time of such a model would be sufficiently small for A2C2 to be able to handle delays.
>
>  Thank you for this insightful observation regarding mode switching. To clarify the capability and design scope of our lightweight head, we categorize "reactivity" into two distinct types. The first type is Local Physical Reactivity, which refers to relatively small changes—such as sensor noise, disturbances, or variations caused by the robot’s own motion—that are not drastic but critically affect accuracy and stability. The correction head is specifically designed for this regime; by leveraging the latest observations, it learns high-frequency residual adjustments to maintain physical consistency (e.g., stabilizing a grasp or tracking a moving target), and since this does not require changing the underlying semantic distribution, a small, fast model (32M parameters, ~4.7 ms inference) is sufficient and effective. The second type is Semantic or Modal Reactivity, which refers to drastic changes in the motion mode itself, such as completely dropping an object or a sudden change in task intent where the distribution mode must change entirely. We agree with your assessment that it is difficult for a lightweight residual head alone to handle these cases, as they require high-level re-planning. However, our system handles this hierarchically: when a such an event occurs, the Asynchronous Base Policy (VLA)—which runs in the background—detects this state change and generates a new action chunk with the correct mode (e.g., "re-grasp") within the inference latency period. Thus, the Correction Head provides Type 1 reactivity to bridge the latency gap, while the Base Policy provides Type 2 reactivity, allowing us to keep the correction head lightweight without requiring it to solve the full mode-switching problem.
>
> > Figure 5 shows the average over all tasks, but a separate comparison within each task would be very important since some tasks in Kinetix require significantly greater closed-loop responsiveness over others. Perhaps this could be done in the appendix? Furthermore, it would be good to show the standard deviation/error within the average result plot.
>
> We appreciate this valuable suggestion. We agree that a task-wise comparison provides important insights, as the requirement for closed-loop responsiveness varies significantly across the Kinetix tasks. We are happy to include the detailed per-task performance breakdown in the Appendix of the final revision to highlight these differences. Furthermore, we will update the average result plots (such as Figure 5) to include standard error bars to clearly illustrate the variance and reliability of our results.

---

> > ### Author Response · Authors · 2025-11-24
> >
> > Regarding the real-world validation, please refer to our General Response where we detail the new experiments on the SO-101 arm showing significant improvements

---

> > > ### Comment · Reviewer_b9NY · 2025-11-26
> > >
> > > Thank you for the detailed response to my comments.
> > >
> > > I do not completely agree with the choice of baselines to evaluate.
> > >
> > > > In the original RTC study, it was demonstrated that RTC consistently outperforms both Temporal Ensembling and BID on these dynamic tasks. Since our results show that A2C2 significantly outperforms RTC (e.g., +23% success rate under delay ), we inferred that A2C2 inherently outperforms the other baselines as well. We omitted them to keep the evaluation focused on the strongest state-of-the-art competitor (RTC).
> > >
> > > I do not think RTC's results can be taken for a universally accepted ranking of baselines. RTC has its own design choices, choice of model to train, choice of environments, etc. and, as such, one cannot simply take the best method according to the RTC paper and compare against that. This is even more so because, as you pointed out in your comment, you have struggled to reproduce the results that RTC reported and, as such, the choice to omit a baseline based off of RTC's reported results is unwise.
> > >
> > > > More importantly, there is a fundamental reason why methods like BID are less suitable for our specific problem setting. While BID uses a smaller policy for guidance, it relies on test-time sampling and optimization, which incurs a high computational cost. In the context of real-time control, this added computation increases the inference latency, potentially worsening the very problem we aim to solve (the "blind spot" during inference).
> > >
> > > Firstly, this is an argument against BID, but not temperal ensembling. Secondly, it's unclear how a small policy being used in BID incurs a high computational cost since the test-time sampling can be parallelized. Note that RTC did not incorporate BID in real-world experiments because BID on a large VLA model can incur latency (not that it's computationally expensive, rather that a forward pass at every time-step in a closed-loop manner just has high latency due to the VLA being a large model), but that argument does not apply to small models. At least for the sake of rigor in experiments and empirical evaluations, these classic baselines should be incorporated at least in the sim experiments.
> > >
> > > > We omitted RTC in the LIBERO Spatial experiments due to significant technical barriers in reproduction.
> > >
> > > RTC has released their code, so you can simply report the results with their implementation. If you do not get the results they have reported, it seems like you should still report the results for rigorous evaluations.
> > >
> > > > As you correctly noted, the Base Policy infers an action chunk only once every steps. Consequently, the latent representation derived from the Base Policy remains identical for all steps within the same action chunk.
> > >
> > > Understood, I think this should be made clearer in the manuscript.
> > >
> > > > However, our central assumption is not that the small correction head learns this complexity from scratch. Instead, our architecture is designed such that the Generalization is guaranteed by the Base Policy. To enable this lightweight design, we feed the Base Policy’s rich latent features directly into the correction head.
> > >
> > > This makes sense to me. I do think that there needs to be rigorous tests on much more complex tasks with VLAs to see if this form of generalization holds in practice, especially when you are just using a small model conditioned on the complex, unstructured latent representations of the VLA.
> > >
> > > Some of my other concerns have been addressed and as such, I am happy to raise my score to a 6. I still think that the choice of baselines is not properly justified and needs to be expanded. Apart from that, my other concern is still regarding how lightweight this additional policy can be and what challenges emerge when using it to edit the action of a significantly larger VLA model -- I think such concerns are best addressed by real-world experiments on significantly more complex tasks than the one in Section 5.3.

---

> > > > ### Author Response · Authors · 2025-12-04
> > > >
> > > > Thank you for your insightful comments.
> > > >
> > > >  Let us elaborate further and provide an update.
> > > >
> > > > >Ensembling and BID on these dynamic tasks. Since our results show that A2C2 significantly outperforms RTC (e.g., +23% success rate under delay ), we inferred that A2C2 inherently outperforms the other baselines as well. We omitted them to keep the evaluation focused on the strongest state-of-the-art competitor (RTC).
> > > > I do not think RTC's results can be taken for a universally accepted ranking of baselines. RTC has its own design choices, choice of model to train, choice of environments, etc. and, as such, one cannot simply take the best method according to the RTC paper and compare against that. This is even more so because, as you pointed out in your comment, you have struggled to reproduce the results that RTC reported and, as such, the choice to omit a baseline based off of RTC's reported results is unwise.
> > > >
> > > > We appreciate the reviewer’s valid point regarding baseline selection. We agree that relying solely on rankings from prior literature is insufficient given the specific design choices and reproduction challenges. To address this, we have explicitly conducted a comparative analysis including Bidirectional Decoding (BID) and Temporal Ensemble (TE) within our experimental setup. As detailed in Appendix 2 and Figure 8, our empirical results confirm that both BID and TE underperform compared to RTC and our proposed method in these environments
> > > >
> > > > > Figure 5 shows the average over all tasks but a separate comparison within each task would be very important since some tasks in Kinetix require significantly greater closed-loop responsiveness over others. Perhaps this could be done in the appendix?
> > > >
> > > > Thanks for pointing out the importance of each task. To see the result of each task, we updated the manuscript, and Appendix 3 and Table 7 show the success rate for each task under an inference delay is 4 and an execution horizon is 4.

---

> > > > > ### Author Response · Authors · 2025-12-04
> > > > >
> > > > > >As you correctly noted, the Base Policy infers an action chunk only once every steps. Consequently, the latent representation derived from the Base Policy remains identical for all steps within the same action chunk.
> > > > > Understood, I think this should be made clearer in the manuscript.
> > > > >
> > > > > We appreciate your understanding and valuable feedback. To improve the clarity of the manuscript, we have revised Section 3.1 to explicitly state this point.
> > > > >
> > > > > > RTC has released their code, so you can simply report the results with their implementation. If you do not get the results they have reported, it seems like you should still report the results for rigorous evaluations.
> > > > >
> > > > > We appreciate your emphasis on rigorous evaluation. While the official RTC codebase for kinetix simulation was available, it did not support VLA architectures at the time of our initial experiments. To address your concern and ensure a fair comparison, we have updated our codebase to incorporate the latest RTC implementation from the LeRobot library. We subsequently evaluated smolVLA with RTC on the LIBERO Spatial benchmark and have reported these new results in Appendix 7.

---

### Official Review · Reviewer_qPXm · 2025-10-25

**Soundness:** 2
**Presentation:** 1
**Contribution:** 3
**Rating:** 2
**Confidence:** 4

**Summary:**

This paper introduces a novel method, Asynchronous Action Chunk Correction (A2C2), to deal with the drawbacks of traditional action chunks, where the chunk length is fixed. Normal action chunks could leave actions out-of-date, because the action really executed in the environment is predicted with observations of several frames ago, and this is not wanted for a reactive behavior to run VLA in a dynamic situation. The proposed A2C2 learns a correction head and it outputs a residual action to refine the original action predicted by the base policy. This kind of method is orthogonal to the real-time chunking. The experiments in Kinetix and LIBERO verify the effectiveness of A2C2.

**Strengths:**

- A2C2 is orthogonal to RTC, bringing new aspects of improving qualities of action chunks, this is interesting.
- The experiments in Kinetix and LIBERO verify the effectiveness of A2C2.

**Weaknesses:**

- Figure 1 could be better for presentation, now it is difficult to understand its meanings. For example, the two emoji faces are not shown anywhere; the color of the emoji is unclear; 4 different action chunks look alike with the only difference being the border color, but what does the color represent?
- A2C2 learns the residual action to fill the gap between ground truth actions and base policy predicted actions. Here the base policy plays a critical role, but there are not adequate ablation studies on the model capacity of the VLA models.
- Learning a residual head for action improvement has been proved effective in many robotics reinforcement learning papers, like PLD or Policy Decorator. What A2C2 has done is to learn a residual refinement head by imitation learning objective, which are alike.
- A2C2 assumes to have access to the training dataset, which is not always possible (for example, one wants to deploy a pretrained VLA model to his environment zero-shot). While RTC is a rule-based refinement technique and plug-and-play, which is more intuitive.

reference:

[1] Self-Improving Vision-Language-Action Models with Data Generation via Residual RL

[2] Policy Decorator: Model-Agnostic Online Refinement for Large Policy Model

**Questions:**

- In the introduction, the author argues that "We first formulated delays in policy inference with VLAs that generate action chunks.". However, RTC also defines $d$ as inference delay. Could the author make this contribution statement more detailed and clear?
- In LIBERO setting, A2C2 uses the latent representation from smolVLM $z_t$. What about training the correction head without this feature? thus making the correction part agnostic to the base policy.
- RTC shows its effectiveness by doing hard tasks in the real world, e.g. strike a match and light the candle. What about the performance of A2C2 in the real world?
- The correction head is indeed a residual head, learning the difference between base policy and the ground truth data. What if using a more advanced VLA model like $\pi_{0.5}$ or X-VLA? I think the residual head works just because of the poor model capacity.

reference:

[1] π0.5: a Vision-Language-Action Model with Open-World Generalization

[2] X-VLA: Soft-Prompted Transformer as Scalable Cross-Embodiment Vision-Language-Action Model

---

> ### Author Response · Authors · 2025-11-21
> **Response to Reviewer qPXm (1/2)**
>
> Thank you for your insightful review. Let me answer your questions.
>
> > Figure 1 could be better for presentation, now it is difficult to understand its meanings. For example, the two emoji faces are not shown anywhere; the color of the emoji is unclear; 4 different action chunks look alike with the only difference being the border color, but what does the color represent?
>
> Thanks for your comment. Let us describe Figure 1 in detail. We intended Figure 1 to illustrate the advantages and disadvantages of asynchronous inference with Action Chunking.
> To answer your specific questions:
> Emojis: The "smiling" face indicates the benefit of continuous control, while the "frowning" face highlights the disadvantage of latency.
> Colors: Different border colors represent different inferred chunks. The same color appears in the sequence to show which chunk the actions belong to.
> Green Blocks: These represent the actions that are actually executed.
> In the specific setting shown ($H=8, e=4, d=2$), the diagram shows that actions are based on observations at least $d$ steps old, deteriorating to $d+e$ steps old at the end of an execution block. We have revised the figure caption to make these meanings explicit.
> Also, as you pointed out, this could be better for presentation, and we are planning to update this figure.
>
> > A2C2 learns the residual action to fill the gap between ground truth actions and base policy predicted actions. Here the base policy plays a critical role, but there are not adequate ablation studies on the model capacity of the VLA models.
>
> We thank the reviewer for this thoughtful comment. We agree that the base policy plays a critical role.
> However, we focused our experiments on SmolVLA because it represented the state-of-the-art at the start of our research. As demonstrated by Shukor et al. (2025), SmolVLA outperforms other recent architectures (such as $\pi_0$) and shows strong real-world performance on standard platforms (e.g., SO-100).
> More importantly, Shukor et al. (2025) (specifically Table 13) highlight that performance degradation over long horizons is a persistent issue even for high-capacity VLAs. Our research focus is specifically on addressing this degradation and handling latency via asynchronous residual correction, rather than evaluating VLA scaling laws. Since comprehensive ablation studies on VLA capacity are available in the original foundational papers (e.g., [Shukor et al., 2025]), we focused our resources on validating the correction mechanism itself.
>
> > Learning a residual head for action improvement has been proved effective in many robotics reinforcement learning papers, like PLD or Policy Decorator. What A2C2 has done is to learn a residual refinement head by imitation learning objective, which are alike.
>
> We thank the reviewer for highlighting the connection to RL residual learning methods like PLD and Policy Decorator. While we share the mathematical formulation of additive residuals ($a_{final} = a_{base} + a_{res}$), our approach differs fundamentally in temporal structure and motivation:
> Asynchronous vs. Synchronous: Existing RL residual methods typically run synchronously, where both the base and residual policies observe the same state $s_t$. In contrast, A2C2 is explicitly designed for asynchronous execution. Our correction head uses the latest observation $s_t$ to correct a base action generated from an outdated observation $s_{t-d}$. A standard synchronous residual head would not solve the latency problem, as it would be blocked by the base policy's inference time.
> Latency vs. Adaptation: The primary motivation of methods like Policy Decorator is domain adaptation or task specialization. Our motivation is latency compensation. We do not aim to change the task behavior, but to align the delayed action with the current environment state.
> Imitation Learning Context: Unlike RL methods that require online interaction and reward engineering, our method operates entirely within the Imitation Learning framework, extracting high-frequency "reflex" behaviors from the same offline dataset used for the VLA.
> If you feel these explanations are not sufficient, I’m happy to answer more points.

---

> > ### Author Response · Authors · 2025-11-21
> > **Response to Reviewer qPXm (2/2)**
> >
> > > A2C2 assumes to have access to the training dataset, which is not always possible (for example, one wants to deploy a pretrained VLA model to their environment, zero-shot). While RTC is a rule-based refinement technique and plug-and-play, which is more intuitive.
> >
> > We respectfully argue that the assumptions of A2C2 and RTC simply target different deployment scenarios:
> > Data Availability (A2C2): In practical robotics, high-performance deployment typically requires fine-tuning the VLA to the specific embodiment or environment. In this standard fine-tuning workflow, access to the training dataset is inherent, making A2C2 easily applicable.
> > Model Architecture (RTC): While RTC is plug-and-play for zero-shot settings, it is architecturally constrained. It relies on the score function or iterative denoising steps of Flow/Diffusion models. It cannot be applied to direct action regression models (like ACT or VQ-Bet) trained with reconstruction loss (e.g., L1), which remain highly popular.
> > Therefore, we view these methods as complementary rather than competing. A2C2 fills the gap for direct regression policies where RTC is inapplicable. Furthermore, for diffusion-based policies, these methods can be combined. In the Appendix 6, we demonstrate that combining A2C2 with RTC yields further performance improvements, confirming their orthogonal benefits.
> >
> > > In the introduction, the author argues that "We first formulated delays in policy inference with VLAs that generate action chunks.". However, RTC is also defines as inference delay. Could the author make this contribution statement more detailed and clear?
> >
> > We agree that RTC also deals with inference delay, but the problem formulation differs significantly in its objective.
> > RTC's Formulation: Focuses on the discontinuity problem caused by delay. It aims to generate smooth transitions between asynchronous chunks using diffusion inpainting.
> > Our Formulation: Focuses on the reactivity problem. We analyze the "age" of the observation used for control. Even if the transition is smooth (RTC), the action may be based on an observation from $d+e$ steps ago.
> > We will refine our contribution statement to clarify that we extend the delay formulation to account for this observation-action latency gap and propose a solution to recover high-frequency reactivity.
> >
> > > In the LIBERO setting, A2C2 uses the latent representation from smolVLM z_t . What about training the correction head without this feature? thus making the correction part agnostic to the base policy.
> >
> > Our design choice is strongly supported by recent research. Kobayashi et al. (2025) demonstrated that even simple architectures (such as ACT) can handle complex multimodal tasks effectively when conditioned on high-quality embeddings like SigLIP. Consistent with this finding, our approach feeds the Base Policy’s rich latent features ($z_t$) directly into the correction head. By leveraging these pre-computed, high-level representations, we allow the correction head to utilize the base model's semantic understanding efficiently. This enables the head to remain lightweight and fast (as shown in Appendix A.5) without sacrificing the performance required for complex language grounding.
> > We do not assume the correction head must be agnostic to the base policy; rather, we view the base policy as a rich feature extractor that the correction head should exploit.
> >
> > https://arxiv.org/pdf/2504.01301
> >
> > > RTC shows its effectiveness by doing hard tasks in the real world, e.g. strike a match and light the candle. What about the performance of A2C2 in the real world?
> >
> > I agree with that real world experiment is really important for robotics paper.
> > Now, we are currently setting up a real-world verification environment to further validate our method's efficacy on physical hardware.
> >
> > > The correction head is indeed a residual head, learning the difference between base policy and the ground truth data. What if using a more advanced VLA model like pi05? I think the residual head works just because of the poor model capacity.
> >
> > First, we selected SmolVLA primarily because it represented the state-of-the-art performance at the time of our experiments. As reported in the original paper (Shukor et al., 2025), SmolVLA outperforms larger models, including $\pi_0$ and OpenVLA, on standard benchmarks despite its efficiency. Furthermore, we chose SmolVLA for its compatibility with the LeRobot framework, which ensures reproducibility and a standardized experimental setup.
> > Also, even if we assume there is a gap in the base model's capacity, we argue that the ability to close this gap using an extremely lightweight correction head (only 32M parameters ) is a significant merit, not a weakness. Scaling up a VLA (e.g., to billions of parameters) to fix marginal errors is computationally expensive and increases latency.

---

> > > ### Author Response · Authors · 2025-11-24
> > >
> > > Regarding the real-world validation, please refer to our General Response where we detail the new experiments on the SO-101 arm showing significant improvements

---

> > > > ### Comment · Reviewer_qPXm · 2025-11-27
> > > > **re**
> > > >
> > > > I have acknowledged the rebuttal.
> > > >
> > > > But my core question has not been well solved, and I still think the residual head works just because of the poor base model performance.
> > > >
> > > > SmolVLA can only achieve 81.8% on LIBERO spatial, while many other SOTA VLA models like $\pi_0$ can reach 96.8%. More simulation experiments on LIBERO with stronger base models and other suites like libero-long, libero-goal, libero-object will help to demonstrate the effectiveness of A2C2.
> > > >
> > > > About the real-world experiments, grasping a cube and placing it in a box is a simple task, thus not enough to demonstrate the effectiveness of A2C2 or even methods like RTC. Because these easy tasks can be done well with current SOTA VLA models. From my experience, a vanilla diffusion policy trained with 100 trajectories of expert-collected data could do this task with a much higher success rate than 38% in Table 2.

---

> > > > > ### Author Response · Authors · 2025-12-04
> > > > >
> > > > > >But my core question has not been well solvedå, and I still think the residual head works just because of the poor base model performance.
> > > > > SmolVLA can only achieve 81.8% on LIBERO spatial, while many other SOTA VLA models like can reach 96.8%. More simulation experiments on LIBERO with stronger base models and other suites like libero# -long, libero-goal, libero-object will help to demonstrate the effectiveness of A2C2.
> > > > >
> > > > > We agree that evaluating on stronger base models is crucial to demonstrate the general applicability of our method. To address this, we have extended our experiments on LIBERO Spatial using Pi0.5, a state-of-the-art VLA model, as detailed in Section 5.2 and Table 2 of the revised manuscript.
> > > > > While we confirmed that it achieves a high success rate of 98.2% with a short execution horizon (e=5), our results highlight that even such a strong model is susceptible to latency and long-horizon execution. Specifically, under a challenging setting with a long execution horizon (e=40) and inference delay (d=10), the performance of the Naïve baseline drops significantly to 82.2%.
> > > > > In contrast, applying A2C2 mitigates this degradation, recovering the success rate to 86.4%. These findings demonstrate that the performance degradation caused by latency and open-loop execution is a fundamental challenge that persists even in high-capacity SOTA models, and A2C2 serves as an effective solution in these scenarios.
> > > > >
> > > > > >About the real-world experiments, grasping a cube and placing it in a box is a simple task, thus not enough to demonstrate the effectiveness of A2C2 or even methods like RTC. Because these easy tasks can be done well with current SOTA VLA models. From my experience, a vanilla diffusion policy trained with 100 trajectories of expert-collected data could do this task with a much higher success rate than 38% in Table 2.
> > > > >
> > > > > We agree with the reviewer that the pick-and-place task is relatively simple and that current SoTA models (or vanilla diffusion policies) can achieve high success rates under standard inference settings (e.g., executing short horizons like 5 steps).
> > > > > However, the low success rate (38%) for the Naïve baseline reported in Table 3  is not due to the model's inability to solve the task, but rather due to the enforced long execution horizon (e=20 and e=40).
> > > > > We intentionally set these long horizons to stress-test the policy under conditions simulating significant inference latency (typical of large VLAs).
> > > > >
> > > > > >Figure 1 could be better for presentation, now it is difficult to understand its meanings. For example, the two emoji faces are not shown anywhere; the color of the emoji is unclear; 4 different action chunks look alike with the only difference being the border color, but what does the color represent?
> > > > >
> > > > > We updated Figure 1. Previously, Figure 1 for an explanation of the problem formulation with asynchronous inference and action chunk. Now, it has been updated to show an overview of our proposed solution. It makes the reader more easy to understand our idea.

---

### Official Review · Reviewer_Scsg · 2025-10-28

**Soundness:** 3
**Presentation:** 1
**Contribution:** 3
**Rating:** 6
**Confidence:** 4

**Summary:**

This paper addresses the critical problem of inference latency in large Vision-Language-Action (VLA) models used for robotic control. The authors identify that the common practice of "action chunking," where a sequence of actions is predicted at once, leads to performance degradation in dynamic environments because the executed actions are based on stale observations. To mitigate this, they propose A2C2 (Asynchronous Action Chunk Correction), a lightweight add-on module that refines the base policy's pre-computed actions in real-time using the most recent observation. A2C2 is trained to predict a residual action, which is added to the base action, effectively closing the loop and providing reactivity without retraining the large, expensive base VLA. The method is evaluated on two distinct benchmarks: the highly dynamic Kinetix suite and the manipulation-focused LIBERO Spatial benchmark. Results demonstrate that A2C2 consistently and significantly improves success rates over naive and other baseline methods (like RTC) under various inference delays and long execution horizons, with minimal computational overhead.

**Strengths:**

1. The paper clearly formulates a critical, real-world problem—the loss of reactivity due to inference latency in large VLAs—that becomes more pressing as models scale. The framing of the problem, including the definitions of delay (d), horizon (H), and execution horizon (e), is precise and insightful.

2. The proposed A2C2 method is conceptually simple, architecturally lightweight, and highly practical. Its design as a plug-and-play residual correction head that can be attached to any pre-trained chunking policy is a major strength, as it avoids the prohibitive cost of fine-tuning large foundation models.

3. The paper provides a thorough empirical validation across two diverse simulation environments (Kinetix and LIBERO), which have different characteristics (highly dynamic vs. spatial reasoning). The experiments systematically test the method under varying delays and horizons, convincingly showing that A2C2 outperforms baselines across the board. The inclusion of inference time benchmarks (showing a 20x speedup over the base policy) solidifies the claim of low overhead.

4. The performance gains are substantial. For instance, on LIBERO with a significant delay (d=10, e=40), A2C2 improves the success rate from 67% to 84%. The results in the Kinetix environment (Table 4) are particularly striking, where A2C2 maintains high success rates even under large delays where other methods fail catasthetically.

5. The authors provide extensive implementation details, hyperparameters in the appendix, and have released source code for both experimental setups, which greatly facilitates reproducibility and future research.

**Weaknesses:**

1. While the core method is validated, a more detailed ablation study would strengthen the paper. For example, the contribution of the specific input components to the correction head (e.g., the temporal feature τ, the base policy's latent representation z, the base action itself) is not thoroughly isolated and analyzed. Understanding which components are most critical would provide deeper insight.

2. The paper focuses on success rates but provides little discussion or qualitative analysis of how the correction head fails when it does, or in what scenarios it might not help. A discussion of the limitations and potential failure modes (e.g., in extremely out-of-distribution states) would provide a more balanced view.

3. The correction head uses different architectures for the two benchmarks (a simple MLP for Kinetix and a more complex transformer encoder for LIBERO). The rationale for this difference, and an ablation showing whether the simpler MLP could suffice for LIBERO (or vice versa), is missing. This leaves some questions about the general architectural requirements for the correction head.

4. The evaluation in LIBERO, while valid, uses a set of instructions that are structurally very similar (all variations of "pick up the black bowl and place it on the plate"). It is unclear how well the method generalizes to more diverse and complex language instructions, which is a key promise of VLAs.

**Questions:**

1. The paper demonstrates A2C2 on a flow-matching policy (Kinetix) and SmolVLA (LIBERO). How sensitive is the performance of the correction head to the type of base policy (e.g., a diffusion policy or a much larger VLA like RT-2)? Is any fine-tuning of the correction head required when switching base models?

2. The method adds a residual to the base action. Were there any observed issues with the correction head producing overly large or unstable corrections that could lead to jerky or unsafe motions, especially at the boundaries between chunks? Is the smoothness purely emergent from the data, or was it explicitly encouraged?

3. The results show that A2C2 helps even in the absence of delay (d=0) on long horizons (e.g., H=50). Does this suggest that the correction head is also compensating for the base policy's own errors in long-horizon open-loop prediction, beyond just compensating for latency? Could A2C2 be seen as a general tool for improving the closed-loop performance of any chunk-based policy?

4. The experiments are in simulation. What are the anticipated biggest challenges in transferring this method to a real-world physical system, where observation noise and dynamics mismatches are significant? Does the correction head risk overfitting to the specific physics of the simulation?

---

> ### Author Response · Authors · 2025-11-21
> **Response to Reviewer Scsg (1/2)**
>
> Thank you for the detailed feedback and close reading!
>
> > While the core method is validated, a more detailed ablation study would strengthen the paper. For example, the contribution of the specific input components to the correction head (e.g., the temporal feature τ, the base policy's latent representation z, the base action itself) is not thoroughly isolated and analyzed. Understanding which components are most critical would provide deeper insight.
>
> We agree with the reviewer that a more detailed ablation study regarding the inputs to the correction head (specifically $\tau$, $z$, and the base action) would greatly strengthen the paper. We acknowledge the importance of verifying which components are most critical for the correction mechanism. We will incorporate an analysis of these architectural choices into the revised manuscript to provide the deeper insight requested.
>
> > The paper focuses on success rates but provides little discussion or qualitative analysis of how the correction head fails when it does, or in what scenarios it might not help. A discussion of the limitations and potential failure modes (e.g., in extremely out-of-distribution states) would provide a more balanced view.
>
>  We appreciate this suggestion. We agree that defining the operating boundaries of the correction head is crucial. We will expand the manuscript to discuss failure modes, specifically noting that while our method effectively handles local disturbances and dynamic inconsistencies (Type 1), it is not designed to solve extreme OoD states (Type 2). We view the handling of extreme OoD scenarios as the role of the robust base policy, while the correction head ensures precision within the valid state distribution.
>
> > The correction head uses different architectures for the two benchmarks (a simple MLP for Kinetix and a more complex transformer encoder for LIBERO). The rationale for this difference, and an ablation showing whether the simpler MLP could suffice for LIBERO (or vice versa), is missing. This leaves some questions about the general architectural requirements for the correction head.
>
>  We thank the reviewer for pointing out this inconsistency. Our primary design philosophy for the correction head was to minimize inference latency for real-time control; therefore, we defaulted to the simplest architecture (MLP) where effective. We utilized a Transformer encoder for LIBERO due to the task's higher complexity and language understanding, but we agree that a direct comparison is missing.
> To address this, we are currently conducting an ablation study on the LIBERO Spatial suite using the simpler MLP architecture. This will allow us to empirically verify whether the additional complexity of the Transformer is strictly necessary or if the lightweight MLP suffices. We will include these results in the revised manuscript to clarify the architectural requirements.
>
> >The evaluation in LIBERO, while valid, uses a set of instructions that are structurally very similar (all variations of "pick up the black bowl and place it on the plate"). It is unclear how well the method generalizes to more diverse and complex language instructions, which is a key promise of VLAs.
>
>  We agree that generalizing to diverse and complex language instructions is a core promise of VLAs.
> However, for this work, we prioritized LIBERO Spatial because our primary contribution lies in the low-level correction of actions and handling spatial perturbations, rather than improving language grounding itself. LIBERO Spatial is a standard benchmark established in the community for evaluating such spatial reasoning and control capabilities. We believe that since our correction head relies on the latent representations of the base policy, better language understanding should naturally follow improvements in the base VLA.

---

> > ### Author Response · Authors · 2025-11-21
> > **Response to Reviewer Scsg (2/2)**
> >
> > > The paper demonstrates A2C2 on a flow-matching policy (Kinetix) and SmolVLA (LIBERO). How sensitive is the performance of the correction head to the type of base policy (e.g., a diffusion policy or a much larger VLA like RT-2)? Is any fine-tuning of the correction head required when switching base models?
> >
> >  This is a valuable point. The performance of the correction head relies on its ability to interpret the specific execution errors of the base policy. Because our method utilizes the base policy's internal features (latents) to infer intent and context, the head is inherently model-specific.
> > Therefore, switching the base policy (e.g., from SmolVLA to RT-2) requires training a new correction head. However, we note that training the correction head is significantly cheaper and faster than training the base policy itself. We consider the investigation of "transferable" correction heads to be an exciting avenue for future work.
> >
> > > The method adds a residual to the base action. Were there any observed issues with the correction head producing overly large or unstable corrections that could lead to jerky or unsafe motions, especially at the boundaries between chunks? Is the smoothness purely emergent from the data, or was it explicitly encouraged?
> >
> >  In our current experimental setup, the smoothness was primarily emergent from the training data, and we did not employ explicit clipping. However, since our method predicts additive residuals, it inherently supports safety clipping. Even if the model were to predict an overly large correction, we can easily cap the magnitude of the residual action during inference to prevent unstable behavior. This is a significant benefit of the residual design for safety-critical applications.
> >
> > > The results show that A2C2 helps even in the absence of delay (d=0) on long horizons (e.g., H=50). Does this suggest that the correction head is also compensating for the base policy's own errors in long-horizon open-loop prediction, beyond just compensating for latency? Could A2C2 be seen as a general tool for improving the closed-loop performance of any chunk-based policy?
> >
> > While our primary motivation was latency mitigation, the performance gain at $d=0$ suggests that A2C2 also compensates for prediction errors and execution drift inherent in open-loop chunk execution. Since standard Action Chunking commits to a trajectory for $H$ steps without feedback, small errors can accumulate even without network latency. Our correction head effectively introduces a closed-loop refinement mechanism, allowing the agent to adjust its action based on the latest observation at every step.
> > Thus, we believe A2C2 has significant potential as a general tool to enhance the precision of chunk-based policies, beyond just latency compensation.
> > > The experiments are in simulation. What are the anticipated biggest challenges in transferring this method to a real-world physical system, where observation noise and dynamics mismatches are significant? Does the correction head risk overfitting to the specific physics of the simulation?
> >
> > We agree that the gap between simulation and reality is a significant challenge. Specifically, we identify Out-of-Distribution (OoD) visual states and actuator noise as the primary hurdles.
> > However, regarding dynamics mismatches, we hypothesize that our method might actually be more robust than standard open-loop chunking. Standard chunking blindly executes a plan for $H$ steps regardless of friction or slippage. In contrast, our correction head introduces closed-loop feedback, allowing the robot to react to unexpected physical dynamics at every step. While overfitting is a risk, we believe that collecting real world data and augmenting it is effective for real-world deployment. We are planning to do the real-world experiments also.

---

> > > ### Author Response · Authors · 2025-11-24
> > >
> > > Regarding the real-world validation, please refer to our General Response where we detail the new experiments on the SO-101 arm showing significant improvements

---

> > > > ### Comment · Reviewer_Scsg · 2025-11-25
> > > >
> > > > I have acknowledged the rebuttal

---

> > > > > ### Author Response · Authors · 2025-12-04
> > > > >
> > > > > Thank you for your insightful comments.
> > > > >
> > > > >  Let us elaborate further and provide an update.
> > > > >
> > > > > >The paper focuses on success rates but provides little discussion or qualitative analysis of how the correction head fails when it does, or in what scenarios it might not help. A discussion of the limitations and potential failure modes (e.g., in extremely out-of-distribution states) would provide a more balanced view.
> > > > >
> > > > > In the updated manuscript, we have added Appendix A.10.5 (Qualitative Analysis for Real World) and Figure 11 to address this concern. Our proposed method enables the robot to maintain a precise trajectory and successfully pick the object, even under long-horizon settings.
> > > > >
> > > > > >While the core method is validated, a more detailed ablation study would strengthen the paper. For example, the contribution of the specific input components to the correction head (e.g., the temporal feature τ, the base policy's latent representation z, the base action itself) is not thoroughly isolated and analyzed. Understanding which components are most critical would provide deeper insight.
> > > > > > The correction head uses different architectures for the two benchmarks (a simple MLP for Kinetix and a more complex transformer encoder for LIBERO). The rationale for this difference, and an ablation showing whether the simpler MLP could suffice for LIBERO (or vice versa), is missing. This leaves some questions about the general architectural requirements for the correction head.
> > > > >
> > > > > For the architectural choice, we did the ablation experiments for the MLP architecture.
> > > > > We appreciate the reviewer's suggestion to clarify the architectural requirements. To address this, we have added Appendix A.6 and Figure 9 in the revised manuscript, where we explicitly conducted an ablation study using a simple MLP architecture for the LIBERO Spatial benchmark.
> > > > > As shown in Figure 9, the simple MLP architecture fails to maintain the base model's performance and yields significantly inferior results compared to our proposed Transformer-based architecture.
> > > > > This result highlights the rationale behind our design choice: while a simple MLP is sufficient for the state-based Kinetix task, it lacks the representational capacity to capture the complex multimodal dependencies (vision and language) required for LIBERO. Therefore, the Transformer encoder is essential for handling high-dimensional multimodal inputs effectively.

---

### Official Review · Reviewer_cxqS · 2025-10-31

**Soundness:** 3
**Presentation:** 2
**Contribution:** 2
**Rating:** 2
**Confidence:** 4

**Summary:**

This paper proposes Asynchronous Action Chunk Correction (A2C2), a lightweight add-on module for vision-language-action (VLA) models that corrects action chunks in real time using the latest observations. The method aims to mitigate performance degradation caused by inference delays and long execution horizons in dynamic robotic tasks. Experiments on Kinetix and LIBERO Spatial benchmarks show improved success rates under varying delays and horizons compared to naive and real-time chunking baselines.

**Strengths:**

1. The work addresses the critical and practical challenge of inference latency in large-scale robotic policies, a key obstacle to real-world deployment.

2. The proposed correction head is modular and can be applied to any pre-trained VLA without retraining, which is appealing for deployment.

**Weaknesses:**

1. No demo support: Without a demo, it is impossible to verify claims about mitigating "jerky motion" or to visually assess the failure modes of baselines. The large quantitative gains would be far more convincing if supported by qualitative evidence showing how A2C2 succeeds where others fail.

2. No real-world experiments: As a robot learning paper, it is difficult to be highly convinced of the method's real-world effectiveness without physical experiments.

If I have misunderstood any points, I am open to discussion.

**Questions:**

1. How does the total inference time (base + correction) compare to the base policy alone? If the correction head runs in parallel, what is the end-to-end latency?

2. The training of $D_{cor}$ uses base policy inferences on the same dataset. Does this lead to overfitting? Have you tested on out-of-distribution tasks or environments?

3. The paper claims the method is “complementary to RTC,” but no experiments combine A2C2 with RTC. Have you tested this combination, and if so, what were the results?

4. Why not directly train a diffusion policy (or other policy) using the ​$D_{cor}$ data format? Such an approach might learn to generate actions conditioned on the latest observations and base actions, potentially solving latency without a separate correction head. Have you explored this? If not, why? If so, how does it compare to A2C2?

5. Why not plug the module in the large-scale VLA (e.g., $\pi_0$ or RDT)? because the latency phenomenon is more pronounced on large VLAs.

---

> ### Author Response · Authors · 2025-11-21
> **Response to Reviewer cxqS**
>
> Thank you for your insightful review.
> As you understand, we addressed a latency issue by adding a correction head. Due to the VLAs' large model, it is challenging to incorporate the latest observations. The correction head solves this problem by asynchronously predicting the residual action.
>
> > No demo support: Without a demo, it is impossible to verify claims about mitigating "jerky motion" or to visually assess the failure modes of baselines. The large quantitative gains would be far more convincing if supported by qualitative evidence showing how A2C2 succeeds where others fail.
>
> To provide the demo, we are planning to do the experiments on the real-world setup. Also, as you said, the decrease in the success rate due to delay and long horizon does not guarantee the “jerky motion”. We will check the video of both the Kinetix and Libero, then we will show some demos.
>
> > No real-world experiments: As a robot learning paper, it is difficult to be highly convinced of the method's real-world effectiveness without physical experiments.
>
> To show the real-world effectiveness of the proposed method. Now, we set up the real-world experiments. I will update as soon as possible.
>
> >How does the total inference time (base + correction) compare to the base policy alone? If the correction head runs in parallel, what is the end-to-end latency?
>
> We assume to use asynchronous inference, and measuring “end-to-end latency” is a bit difficult because the base and correction head run in parallel.  But the inference time of our proposed methods is about 5ms (mentioned in the Appendix Table 8). Using asynchronous inference with an action chunk, the robot no longer pauses due to the base policy’s inference time. However, the initial parts of each action chunk cannot be used, and all executed actions are always inferred from observations that are delayed. Our proposed method tackles this problem, and the inference time needs to be smaller than 1 step. If I have misunderstood your point, I would greatly appreciate it if you could let me know.
>
> > The training of D_corr  uses base policy inferences on the same dataset. Does this lead to overfitting? Have you tested on out-of-distribution tasks or environments?
>
> We haven’t tried the OoD tasks or environments yet due to the limitations of the simulation environment. As you said in the weakness, we need to think about the real-world experiments and plan to do so.
>
>
> >The paper claims the method is “complementary to RTC,” but no experiments combine A2C2 with RTC. Have you tested this combination, and if so, what were the results?
>
> Thanks for the insightful questions. With RTC and our proposed method, it shows better results. Please see Appendix 6.
> These results indicate that our proposed method is complementary to RTC, and the two can be jointly applied.
>
> > Why not directly train a diffusion policy (or other policy) using the ​
>  data format? Such an approach might learn to generate actions conditioned on the latest observations and base actions, potentially solving latency without a separate correction head. Have you explored this? If not, why? If so, how does it compare to A2C2?
>
> Using a diffusion policy or ACT alone would not directly resolve the latency issue without a separate correction head. Our data format explicitly assumes access to the base policy’s action chunk, and during inference, we must still run the base policy to produce the chunk. This makes it difficult for a single diffusion-style model to incorporate the most recent observation in real time, because generating the full action sequence with the base policy remains the bottleneck.
> If your point is instead about replacing our correction head with a diffusion or ACT-style model, that is a reasonable architectural alternative. In fact, the correction head we experimented with in the LIBERO Spatial setting is partly inspired by ACT. However, we intentionally keep the correction module small: a lightweight encoder followed by an MLP that predicts a residual added to the base action. This design allows the correction to run fast enough to compensate for the latency of the base policy.
> If I have misunderstood your intended comparison—whether you meant training a unified policy from scratch, replacing only the correction head, or questioning the data format itself—I would appreciate it if you could clarify.
>
> > Why not plug the module in the large-scale VLA (e.g., pi_0 or RDT)? because the latency phenomenon is more pronounced on large VLAs.
>
> We selected SmolVLA as the base policy for several practical and scientific reasons. First, at the time we began our experiments, SmolVLA demonstrated state-of-the-art performance; according to Shukor et al. (2025), it outperforms π0 and OpenVLA. Second, SmolVLA is fully integrated into LeRobot, which makes controlled and reproducible experimentation substantially easier. These factors made it a natural starting point, and we expect our method to apply to larger-scale VLAs as well.

---

> > ### Author Response · Authors · 2025-11-24
> >
> > Regarding the real-world validation, please refer to our General Response where we detail the new experiments on the SO-101 arm showing significant improvements

---

> > > ### Comment · Reviewer_cxqS · 2025-11-27
> > >
> > > Due to the author's efforts, I increased my score from 2 to 4.

---

> > > > ### Author Response · Authors · 2025-12-04
> > > >
> > > > Thank you for your insightful comments.
> > > >
> > > >  Let us elaborate further and provide an update.
> > > >
> > > > > The training of uses base policy inferences on the same dataset. Does this lead to overfitting? Have you tested on out-of-distribution tasks or environments?
> > > >
> > > > We did not do the OoD setting in the simulator. However, our real-world experiments demonstrate the model's robustness to spatial variations. As detailed in Appendix A.10.2, during the data collection for training, we repositioned the cubes and the bowl randomly in every episode to ensure generalization.
> > > > In contrast, for the evaluation phase described in Appendix A.10.4, we placed the objects in fixed positions to ensure a fair and consistent comparison across different conditions. The fact that the model, trained on randomized positions, could successfully perform the task in the evaluation setting demonstrates that the correction head learns to generalize rather than overfit to specific training trajectories.
> > > >
> > > > >Why not plug the module in the large-scale VLA (e.g.,  or RDT)? because the latency phenomenon is more pronounced on large VLAs.
> > > >
> > > > We agree with the reviewer that inference latency is a more pronounced issue for large-scale VLAs. To address this, we have extended our evaluation to include Pi0.5, a state-of-the-art large VLA model.
> > > > As shown in our new Appendix A.9, the inference time for Pi0.5 is approximately 197 ms, which is nearly double that of the SmolVLA baseline (101 ms). This confirms that the latency bottleneck is indeed critical for larger models.
> > > > In Table 2, we demonstrate that while Pi0.5 is a strong baseline, its performance still degrades to 82.2% under long execution horizons (e=40) and injected delay (d=10). Crucially, applying our A2C2 module mitigates this degradation, recovering the success rate to 86.4%. These results confirm that our method is effective and necessary even for high-capacity, large-scale VLA models where latency is a significant factor.

---

### Author Response · Authors · 2025-11-24
**New Real-World Experiments Validation**

We thank the reviewer for the constructive feedback regarding real-world validation. We agree that demonstrating the method's effectiveness on physical hardware is crucial.

To address this, we have conducted additional real-world experiments using an SO-101 robotic arm. We have added a new Section 5.3 (Real World Task) and Appendix A.7 to the revised paper.
We evaluated a "pick-and-place" task where the robot must grasp a target object and place it into a container. : As shown in Table 2, A2C2 significantly outperforms the baseline, especially at long horizons. For instance, at an execution horizon of $e=20$, A2C2 achieved an 84% success rate compared to only 38% for the Naïve baseline3. Even at $e=40$, A2C2 maintained a 58% success rate while the baseline dropped to 26%.

These results confirm that our proposed lightweight correction head is effective not only in simulation but also in real-world environments with physical latency and dynamics.

Finally, we also plan to conduct validation in multi-task settings and under delayed environments in our real-world experiments.

---

### Author Response · Authors · 2025-12-04

Dear Program Chairs, Senior Area Chairs, Area Chairs, and Reviewers,

 We sincerely appreciate the constructive feedback and expert insights. We have carefully addressed each concern, and the revised manuscript now includes substantial updates. As the discussion phase concludes, we present a concise summary of the key revisions and newly added experiments introduced during the rebuttal period to directly resolve the core issues raised.Please note that all significant changes in the revised manuscript have been marked in red for easy reference.

1. No Real World Experiment. (Reviewers cxqS, qPXm)

To address concerns regarding the lack of real-world validation, we conducted additional experiments using the SO-101 robot arm. Our results demonstrate that while the base policy fails to grasp the cube under long-horizon conditions, our correction head successfully rectifies the action to achieve a stable grasp. Please refer to Appendix 10 for details.

2. Using larger and SOTA VLA models as base models. (Reviewers cxqS,qPXm)

To demonstrate the generalizability of our method to state-of-the-art VLA models, we conducted additional experiments using pi05. As shown in Table 2, while the performance of the standalone pi05 model on LIBERO Spatial degrades with longer horizons and inference delays, our method maintains robust performance. This indicates that our problem formulation is effective even for SOTA VLAs and scales well to larger, more advanced models.

3. Integrate RTC and our proposed method. (reviewer cxqS, Scsg)

While our problem formulation aligns with that of RTC, the underlying methodologies are fundamentally different, as clarified in Section 5.1. This distinction allows for the seamless integration of RTC with our A2C2. As demonstrated in Figure 5, this combination yields the highest performance, outperforming individual methods.

4. Qualitative analysis (reviewer cxqS)

To show how our correction head works, we added qualitative analysis for real-world experiments. Please see Appendix 10 and Figure 11 for details.

5. Comparison with RTC on LIBERO Spatial (reviewer b9NY)

While the official RTC codebase for kinetix simulation was available, it did not support VLA architectures at the time of our initial experiments. To ensure a fair comparison, we have updated our codebase to incorporate the latest RTC implementation from the LeRobot library. We subsequently evaluated smolVLA with RTC on the LIBERO Spatial benchmark and have reported these new results in Appendix 7.

6. Discussion for Model architecture design. (reviewer Scsg)

For the architectural choice, we did the ablation experiments for mlp architecture with LIBERO Spatial. We appreciate the reviewer's suggestion to clarify the architectural requirements. To address this, we have added Appendix A.6 and Figure 9 in the revised manuscript, where we explicitly conducted an ablation study using a simple MLP architecture for the LIBERO Spatial benchmark.
As shown in Figure 9, the simple MLP architecture fails to maintain the base model's performance and yields significantly inferior results compared to our proposed Transformer-based architecture.
This result highlights the rationale behind our design choice: while a simple MLP is sufficient for the state-based Kinetix task, it lacks the representational capacity to capture the complex multimodal dependencies (vision and language) required for LIBERO. Therefore, the Transformer encoder is essential for handling high-dimensional multimodal inputs effectively.

7. BID and Temporal Ensemble comparison. (reviewer b9NY)

To compare with more baselines, we have explicitly conducted a comparative analysis including Bidirectional Decoding (BID) and Temporal Ensemble (TE) within our experimental setup. As detailed in Appendix 2 and Figure 8, our empirical results confirm that both BID and TE underperform compared to RTC and our proposed method in these environments.

8. Update Figure 1 for better clarity and easier understanding. (reviwer, qPXm)

We addressed the reviewers' concerns regarding the paper's presentation and the clarity of Figure 1. We acknowledged that the original figure, which focused on the problem formulation (specifically, asynchronous inference and action chunking), was not optimal for a general introduction. Therefore, we have redesigned Figure 1 to instead illustrate the high-level overview of our proposed solution. We believe this update significantly improves readability and makes it easier for readers to grasp the core concept of our approach.

---

### Meta-Review · Area_Chair_DGEj · 2026-01-02

**Summary:**

The submission addresses the problem of execution degradation of robot manipulation policies due to inference and communication latencies. The proposed solution is A2C2, a method for learning a residual policy that has access to the most recent observations at all times and applies a correction to each action in the action chunk returned by the base policy.

The initial set of reviews acknowledge the importance of the problem but raised a number of concerns, the most important ones, in the meta-reviewer's opinion, being:
- The lack of evaluation on physical robots and tasks.
- The lack of evaluation on strong VLAs.
- The lack of evaluation of A2C2+RTC.

It should be noted that the reviewers raised other concerns too, including requests for ablation studies, qualitative empirical analysis, and the residual model architecture explanations.

During the rebuttal period, the authors have added a physical-robot evaluation on a SO-101 robot on a cube pick-and-place task, included Pi0.5 into the LIBERO-Spatial evaluation, and evaluated a combination of A2C2+RTC on the Kinetix environment.

The metareviewer thinks that, despite the additional experiments, the empirical evaluation is still inadequately convincing and proposes the following modifications to improve it in the future iterations:

- **Include Pi0.5+RTC as a baseline for all evaluation environments.** This is currently the best-performing VLA with real-time execution adaptation, and it looks strange that it's used only in the LIBERO-Spatial experiments in the current paper version. The metareviewer understands that this is likely due to a LIBERO-finetuned Pi0.5 checkpoint being available, but it's not a sufficient reason to omit the results on the other environments from the paper, especially given the very long rebuttal period this year.

- **Do experiments on a real task more dynamic than cube pick-and-place.** The physical experiments currently look limited and contrived: cube pick-and-place is a very simple task, and the experiment setup artificially complicates it by using SmolVLA with a very large execution horizon. It's evident from the LIBERO-Spatial results on Pi0.5 and SmolVLA in Tables 1 and 2 that SmolVLA's policies scale much more poorly with execution horizon than Pi0.5's, which is likely the reason why A2C2 produces much smaller, if at all significant, lift for Pi0.5 on LIBERO-Spatial. The same can be expected in the physical experiments. Also, it's difficult to find a justification for using the execution horizon as large as 40.

- **Provide the quantitative the results on the physical task.** What's the success rate comparison? Neither Section 5.3 nor Appendix 10 of the revised manuscript that the authors are referring to in the rebuttal has these numbers.

- **Provide confidence intervals for the success rates on the simulated tasks.** For many of the comparisons on Kinetix and LIBERO-Spatial, it's very non-obvious whether they are statistically significant, e.g., 90.6% vs 92.4% or 91.2% vs 92% in Table 2.

**Reviewer Concerns:**

The concerns regarding explanations of the A2C2 architecture and qualitative empirical analysis have been addressed sufficiently. The additional experiments using Pi0.5 and RTC have been illuminating but insufficient (please see the Summary). The physical experiments need significant improvement (please see the Summary for details).

**Reviewer Scores:**

Reviewer cxqS explicitly stated that they would have increased the score from 2 to 4 and reviewer b9NY from 4 to 6. The other two reviewers would have probably left their scores as is (2 and 6), judging by their responses to the rebuttals.

---

### Decision · Program_Chairs · 2026-01-26

Reject